

# Efficient and flexible approach to simulate low-dimensional quantum lattice models with large local Hilbert spaces

**Thomas Köhler[1], Jan Stolpp[2] and Sebastian Paeckel[3⋆]**

**1** Department of Physics and Astronomy, Uppsala University,
Box 516, S-751 20 Uppsala, Sweden
**2** Institut für Theoretische Physik, Georg-August-Universität Göttingen,
37077 Göttingen, Germany
**3** Department of Physics, Arnold Sommerfeld Center for Theoretical Physics (ASC),
Munich Center for Quantum Science and Technology (MCQST),
Ludwig-Maximilians-Universität München, 80333 München, Germany

⋆ sebastian.paeckel@physik.uni-muenchen.de

## Abstract

Quantum lattice models with large local Hilbert spaces emerge across various fields in quantum many-body physics. Problems such as the interplay between fermions and phonons, the BCS-BEC crossover of interacting bosons, or decoherence in quantum simulators have been extensively studied both theoretically and experimentally. In recent years, tensor network methods have become one of the most successful tools to treat such lattice systems numerically. Nevertheless, systems with large local Hilbert spaces remain challenging. Here, we introduce a mapping that allows to construct artificial $U(1)$ symmetries for any type of lattice model. Exploiting the generated symmetries, numerical expenses that are related to the local degrees of freedom decrease significantly. This allows for an efficient treatment of systems with large local dimensions. Further exploring this mapping, we reveal an intimate connection between the Schmidt values of the corresponding matrix-product-state representation and the single-site reduced density matrix. Our findings motivate an intuitive physical picture of the truncations occurring in typical algorithms and we give bounds on the numerical complexity in comparison to standard methods that do not exploit such artificial symmetries. We demonstrate this new mapping, provide an implementation recipe for an existing code, and perform example calculations for the Holstein model at half filling. We studied systems with a very large number of lattice sites up to $L = 501$ while accounting for $N_{\mathrm{ph}} = 63$ phonons per site with high precision in the CDW phase.



# 1  Introduction

Large local Hilbert spaces appear in various kinds of problems in quantum many-body physics. Prominent examples arise in the field of ultra-cold quantum gases. Systems such as interacting bosons in a one-dimensional lattice [1, 2] or trapped ion quantum simulators [3–5] have been studied extensively, fertilizing a rapid theoretical and experimental progress. Another typical problem featuring large local Hilbert spaces is the interplay between lattice fermions and phonons. For instance, the formation and stability of (Bi-)Polarons is a central problem and considerable effort has been taken for its investigation [6–15]. A broad class of different methods such as quantum Monte Carlo [16, 17], density-functional theory [18], density-matrix embedding theory [19], or dynamical mean-field theory [20–22] has been explored to study its various aspects. Evidently, the task to numerically describe such low-dimensional, strongly-correlated quantum systems has been subject to a vast development. In particular, the capabilities of tensor-network methods have improved a lot in the past two decades. Here, matrix-product states (MPSs) have become the fundament for flexible, numerically unbiased and in principle exact methods allowing for the study of not only ground-state properties but also of out-of-equilibrium dynamics of quantum many-body systems [23–33].

In (time-dependent) density-matrix renormalization group (DMRG) methods, the computationally limiting factor is the bond dimension of the tensors when performing tensor contractions [26–28, 30, 31, 33]. For instance, using MPS, one is mostly concerned with matrix-matrix contractions, which scale with the third power of the dimensions of the involved matrices. However, these operations can be rendered cheaper if the system under consideration conserves global symmetries. Being able to exploit (non-)abelean symmetries is an important feature of tensor networks in general [34–37], as a large bond dimension is related to the amount of entanglement and decay of correlation functions [29, 38, 39]. Aiming to describe strongly correlated systems, large bond dimensions can be required and thereby exploiting as

many symmetries of the system as possible is highly desired.

Another important contribution to the numerical expenses of MPS algorithms is the dimension of the local Hilbert spaces $\mathcal{H}_j$. For instance, when considering systems with large spin or bosonic degrees of freedom, a local dimension $\dim \mathcal{H}_j \equiv d_j \sim \mathcal{O}(10) - \mathcal{O}(100)$ can yield drastic restrictions on the maximum possible bond dimensions as typical contractions usually scale with $d_j^2$ or even $d_j^3$. In order to overcome such restrictions, approaches such as the pseudo site (PS) and the local-basis optimization (LBO) method [6, 40–42] were developed. These methods have proven to be successful tools for treating fermion-phonon couplings in the Holstein model, even out of equilibrium and at finite temperature [6, 10, 13, 14].

In this paper, we introduce an alternative approach to simulate systems with large local Hilbert spaces efficiently and in a flexible framework. In order to treat these kinds of systems efficiently with MPS, we exploit the fact that global $U(1)$ symmetries reduce effective local block dimensions drastically [34–36]. The starting point of our method is a thermofield doubling of the many-body Hilbert space, which is an established procedure in finite-temperature DMRG [43–45]. Then, introducing a mapping for operators into a particular subspace of the doubled Hilbert space allows us to show that global operators breaking $U(1)$ symmetries can be identified with *projected purified operators* that conserve the corresponding symmetries[1]. Thereby, challenging general lattice systems breaking global $U(1)$ symmetries with $d_j \sim \mathcal{O}(10) - \mathcal{O}(100)$ can always be mapped into numerically more feasible systems. Importantly, this mapping requires only minor changes in existing codes and is completely general.

The paper is organized as follows: At first, we present the relevant aspects of our approach in Sec. 2 in a less detailed fashion and provide an implementation recipe, in Sec. 3, which captures the changes in actual codes. In Sec. 4, we introduce the *projected purification* in great detail and show how to construct corresponding operators. In Secs. 5 and 6, we present the representation in terms of MPS and discuss the connection between the Schmidt values on the newly emerging auxiliary bonds and the diagonal elements of the Single-Site Reduced Density Matrix (1RDM). We illustrate our mapping in Sec. 7 with an exemplary application of our mapping to the Holstein model and present numerical results demonstrating its computational capabilities. Finally, we conclude and discuss further applications in Sec. 8. Additional technical details related to both, the method and applications can be found in the appendices.

## 2 General Concept

The general idea of our mapping is to exploit global $U(1)$ symmetries, where the system under consideration does not conserve them in the first place. In the tensor-network framework, states can be constructed so that they transform under a global symmetry, i.e., they are eigenstates of the corresponding symmetry generator. Let us consider a system with a global particle number operator $\hat{N}$ that, for now, is not a conserved quantity of the system. Eigenstates $|N\rangle$ of $\hat{N}$ are labeled by their irreducible representations $N$ and (ignoring degeneracies) any state can be decomposed in terms of these eigenstates

$$|\psi\rangle = \sum_N \psi_N |N\rangle \ . \tag{1}$$

Now, we can perform a doubling of the original Hilbert space and construct states of the form

$$|\psi\rangle_{PB} = \sum_{N,N'} \psi_{N,N'} |N\rangle_P \otimes |N'\rangle_B \ , \tag{2}$$

---

[1]A side note: The construction is closely related to the formulation of supersymmetry in high-energy physics. Even though, supersymmetry itself is not possible for lattice systems by construction, the general prescriptions of our method show striking similarities [46].

where we introduced labels $P, B$ to distinguish the different Hilbert spaces. We restrict the allowed coefficients $N'$ such that each state $|N\rangle$ can be mapped uniquely to a state $|N\rangle_P \otimes |N_0 - N\rangle_B$ with a properly chosen $N_0$:

$$|N\rangle \longmapsto |N\rangle_P \otimes |N_0 - N\rangle_B \ . \tag{3}$$

The transformed wavefunctions

$$|\psi\rangle \longmapsto \sum_N \psi_{N,N_0} |N\rangle_P \otimes |N - N_0\rangle_B \ , \tag{4}$$

are eigenstates of the new, global symmetry $\hat{N}_P + \hat{N}_B$ with eigenvalue $N_0$ and can therefore be represented efficiently by symmetric MPS. The subspace $\mathcal{P}$ spanned by all states $|N\rangle_P \otimes |N_0 - N\rangle_B$ has the same dimension as the original Hilbert space so that no additional complexity is generated with this new representation.

An important observation is that the coefficients $\psi_{N,N_0}$ can be recast into a block-$N \times N$ matrix $\psi_{N,N'}$ and each block can be factorized using a singular-value decomposition (SVD)

$$
\begin{aligned}
|\psi\rangle \equiv \quad \sum_N \psi_{N,N_0} |N\rangle_P \otimes |N - N_0\rangle_B &= \sum_{N,N'} \psi_{N,N'} \delta(N' - (N - N_0)) |N\rangle_P \otimes |N'\rangle_B \\
&= \sum_N \Lambda_N \psi_{P;N} |N\rangle_P \otimes \psi_{B;N} |N - N_0\rangle_B \ .
\end{aligned} \tag{5}
$$

Here, $\psi_{P/B;N}$ are left-/right-orthonormal matrices that are obtained by factorizing the degenerated blocks $\psi_{N,N_0}$ for fixed $N$ and $\Lambda_N$ are diagonal matrices. Normalization of the overall state demands $\sum_N \text{Tr} \Lambda_N^2 = 1$ so that

$$\hat{\rho} = \text{Tr}_B |\psi\rangle \langle\psi| = \sum_N \psi_{P;N} \Lambda_N^2 \psi_{P;N}^\dagger |N\rangle_P {}_P\langle N| \ , \tag{6}$$

is a density operator, which describes the mixture of the different irreducible representations labeled by $N$. Note that $\rho_N = \text{Tr} \Lambda_N^2$, i.e., the diagonal elements of $\hat{\rho}$, specify the mixing of symmetry sectors in the state $|\psi\rangle$. As an example consider a nearly $U(1)$-symmetry conserving state that is characterized by a dominating diagonal element $\rho_N \approx 1$. The remaining, quickly decaying elements $\rho_N$ allow us to truncate the state representation so that a compression scheme in the subspace $\mathcal{P}$ can be formulated, which is in complete accordance to the canoncial truncation scheme used in DMRG. Importantly, the same considerations can be applied to the local degrees of freedom, constituting the many-body Hilbert space.

Guided by this idea we show in the following sections that there is a simple prescription to transform operators so that they are acting in $\mathcal{P}$ only. Using balancing operators $\hat{\beta}_{B;j}^{(\dagger)}$ (which are introduced in Eqs. (19) and (20)), global operators $\hat{O}$ that break the global $U(1)$ symmetry generated by $\hat{N}$ can be mapped into operators conserving the global $U(1)$ symmetry generated by $\hat{N}_P + \hat{N}_B$. This is achieved by replacing ladder operators $\hat{b}_j^{(\dagger)}$ in the original Hilbert space:

$$
\begin{aligned}
\hat{b}_j &\longmapsto \hat{b}_{P;j} \otimes \hat{\beta}_{B;j}^\dagger \\
\hat{b}_j^\dagger &\longmapsto \hat{b}_{P;j}^\dagger \otimes \hat{\beta}_{B;j} \ .
\end{aligned} \tag{7}
$$

The detailed mapping, containing also the intermediate step of doubling the Hilbert space, is shown in Fig. 1. Note that our mapping is also valid for fermionic degrees of freedom, e.g., electrons with a pairing term that breaks $U(1)$ symmetry. Nevertheless, the general definition of the bosonic balancing operators $\hat{\beta}_{B;j}^{(\dagger)}$ remains unchanged even in this case.

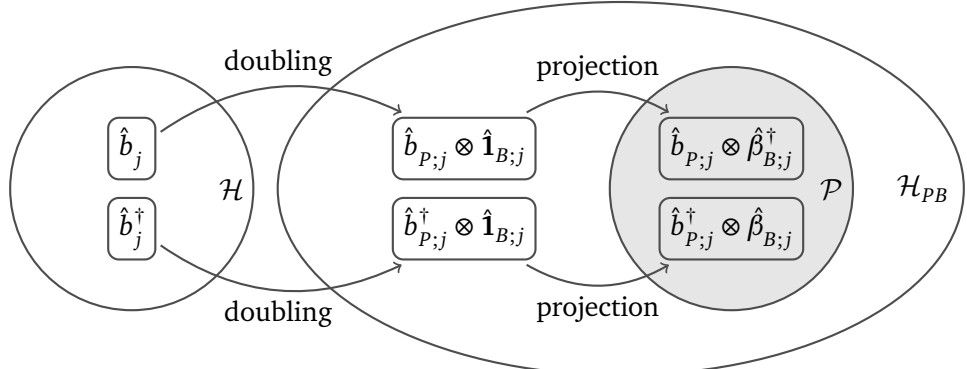

Figure 1: Mapping of local operators $\hat{b}_j$ and $\hat{b}_j^\dagger$ acting on $\mathcal{H}$ into projected purified local operators $\hat{b}_{P;j}\hat{\beta}_{B;j}^\dagger$ and $\hat{b}_{P;j}^\dagger\hat{\beta}_{B;j}$ acting on $\mathcal{P}$. This transformation is the central, necessary modification for existing codes in order to use our method.

Recapitulating this short description of the general ideas of our mapping it should be noted that the states mapped to $\mathcal{P}$ are pure states in $\mathcal{P}$ but describe mixed states with respect to the orthogonal decomposition of $\mathcal{H}$ in terms of the eigenstates of $\hat{N}$. This is in close reminiscence to the purification procedure [43–45] that is commonly used to represent mixed states with respect to $\mathcal{H}$. However, there is also an important difference: Restricting the allowed states by a projection into the subspace $\mathcal{P}$ of the doubled Hilbert space, the complexity of the state's representation is conserved, i.e., our mapping does not add additional degrees of freedom to the problem under consideration.

## 3 Implementation Recipe

Next, we provide a short recipe, for how to implement the previously described projected purified DMRG (ppDMRG) for ground-state searches and time-evolution methods, including prerequirements. Note that this recipe is particularly short, because the necessary changes are small.

**Prerequirements**   In order to incorporate ppDMRG into an existing framework, it is necessary that the framework can handle Hamiltonians with more than nearest-neighbor interactions.

**Necessary changes**   The existing set of local operators needs to be extended with balancing operators that act on the bath sites, as introduced in Eqs. (19) and (20). In particular, for every species of local creation and annihilation operators corresponding balancing operators are needed when changing a global $U(1)$ quantum number. Those operators shall only have zero and one as elements and always commute with every other operator. Additionally, for each species of creation- and annihilation operators $\hat{b}_j^{(\dagger)}$, a parity-operator $\hat{P}_{\hat{b}_j} = e^{i\pi\hat{b}_j^\dagger\hat{b}_j}$ might be useful. A scenario in which the action of $\hat{P}_{\hat{b}_j}$ is necessary is discussed in App. C.

**Usage**   Following these changes, all existing tools can be used as usual, but with a doubled system size where physical and bath sites alternate, which is a common technique in finite-temperature DMRG. Hence, local observables are now evaluated via two neighboring operators. Note that there is no need to map the state back into the original Hilbert space since

the physical and the original Hilbert space are isomorphic to each other, as we show in Sec. 4. However, care must be taken that the MPS represents states in $\mathcal{P}$, i.e., the $L$ local gauge constraints defined in Eq. (18) have to be fulfilled. Fortunately, since projected purified operators manifestly act on $\mathcal{P}$ only, it suffices to ensure that the initial state of any algorithm is in $\mathcal{P}$. For instance, using the previous conventions, an initial state for a ground-state search is given by the product state

$$|\psi\rangle = |n_{P;1} = 0\rangle \otimes |n_{B;1} = \sigma - 1\rangle \otimes \cdots |n_{P;L} = 0\rangle \otimes |n_{B;L} = \sigma - 1\rangle \ . \tag{8}$$

Clearly, for typical ground-state calculations this state is a bad initial guess. However, it can be used as a starting point to create more suitable initial guess states by applying sequences of projected purified operators. Additionally, our numerical experiences gained so far suggest that the convergence of ground state calculations can benefit from a careful use of the subspace expansion [47].

## 4  General Models and Bath Sites

We consider a lattice system of $L \in \mathbb{N}$ degrees of freedom, each of which being described within a Hilbert space $\mathcal{H}_\sigma$ of local dimension $\sigma \in \mathbb{N}$ spanning the system's overall tensor-product Hilbert space $\mathcal{H} = \mathcal{H}_\sigma^{\otimes L}$. A state $|\psi\rangle \in \mathcal{H}$ can be expressed in terms of all local degrees of freedom $|\sigma_1 \cdots \sigma_L\rangle \in \mathcal{H}$:

$$|\psi\rangle = \sum_{\sigma_1,\ldots,\sigma_L} \psi_{\sigma_1 \ldots \sigma_L} |\sigma_1,\ldots,\sigma_L\rangle \ , \tag{9}$$

with, in general, complex coefficients $\psi_{\sigma_1 \ldots \sigma_L} \in \mathbb{C}$.

Let $\hat{O}$ be an operator acting on this tensor product Hilbert space $\mathcal{H}$ and let $\hat{N} = \sum_j \hat{n}_j$ be another operator with local operators $\hat{n}_j : \mathcal{H}_\sigma \longmapsto \mathcal{H}_\sigma$ fulfilling the commutation relations $\left[\hat{n}_j, \hat{n}_k\right] = 0$. We denote the ladder operators spanning the algebra of local operators by $\hat{b}_j^{(\dagger)}$ that obey canonical commutation relations $\left[\hat{b}_j, \hat{b}_k^\dagger\right]_\epsilon = \delta_{j,k}$ and $\epsilon = \pm$ distinguishes between the commutator or anticommutator. Without loss of generality, we choose the spectrum of the local operators $\hat{n}_j$ to be $n_j \in \{0, 1, \ldots, \sigma - 1\}$[2]. Let us assume furthermore that $\hat{O}$ contains summands with ladder operators $\hat{b}_j^{(\dagger)}$ that are not paired up with their Hermitian conjugates breaking the global $U(1)$ symmetry generated by $\hat{N}$. For instance, in the Holstein model (see Sec. 7) such contributions are given by the fermion-phonon interactions

$$\hat{O} = -\sum_j \hat{n}_j^f \left(\hat{b}_j^\dagger + \hat{b}_j\right) \quad \Rightarrow \quad \left[\hat{N}, \hat{O}\right] \neq 0 \ . \tag{10}$$

Note that in this example $\hat{N} = \sum_j \hat{b}_j^\dagger \hat{b}_j$ is the operator counting the number of phonons and $\hat{n}_j^f$ measures the local fermion density.

Next, we introduce a thermofield doubling of this Hilbert space. The new double Hilbert space $\mathcal{H}_{PB} = \mathcal{H}_P \otimes \mathcal{H}_B$ consists of two copies of the original Hilbert space, which we denote as the physical Hilbert space $\mathcal{H}_P$ and the bath Hilbert space $\mathcal{H}_B$ (see first arrow in Fig. 2). Correspondingly, we denote the density operators $\hat{n}_{P;j}$ and $\hat{n}_{B;j}$, which have exactly the same properties as the density operators $\hat{n}_j$ in the original Hilbert space. In particular, the basis states $|n_{P/B;1}\rangle \otimes \cdots \otimes |n_{P/B;L}\rangle \equiv |n_{P/B;1} \cdots n_{P/B;L}\rangle$ span a complete orthonormal basis of $\mathcal{H}_{P/B}$.

---

[2]In fact, the following discussion is valid for any labeling of the irreducible representations of the $U(1)$ symmetries.

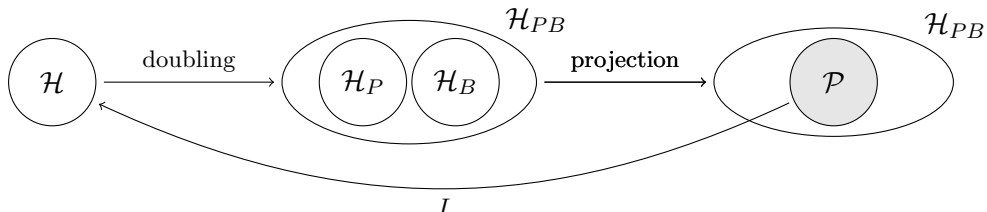

Figure 2: Starting from some Hilbert space $\mathcal{H}$, a thermofield doubling is performed to obtain the combined Hilbert space $\mathcal{H}_{PB} = \mathcal{H}_P \otimes \mathcal{H}_B$. Applying the projection as discussed in the main text yields the subspace $\mathcal{P}$, in which the global $U(1)$ symmetry is restored. Finally, upon acting with $I$ as introduced in Eq. (14), states in $\mathcal{P}$ are identified with states in $\mathcal{H}$.

Here, we leave the framework of finite-temperature DMRG by considering the subspace $\mathcal{P} \subset \mathcal{H}_{PB} = \mathcal{H}_P \otimes \mathcal{H}_B$ of the doubled system that is spanned by all states

$$\left| n_{P;1}, \ldots, n_{P;L} \right) = \left| n_{P;1}, \ldots, n_{P;L} \right\rangle_P \otimes \left| g(n_{P;1}), \ldots, g(n_{P;L}) \right\rangle_B \tag{11}$$

$$= \left| n_{P;1}, \ldots, n_{P;L}, g(n_{P;1}), \ldots, g(n_{P;L}) \right\rangle_{PB}, \tag{12}$$

with $n_{P;j} \in [0, \sigma - 1]$ and $g(x) = \sigma - 1 - x$ (see second arrow in Fig. 2). Note that for convenience we have labeled the kets in the physical and bath system by subscripts and introduced rounded kets to indicate states in the subspace $\mathcal{P} \subset \mathcal{H}_{PB}$, which depend only on a reduced number of coefficients $n_{P;1}, \ldots, n_{P;L}$. This subspace is contained in the subspace with $N_P + N_B = (\sigma - 1) \cdot L$, i.e.,

$$(\hat{N}_P + \hat{N}_B) \left| n_{P;1}, \ldots, n_{P;L} \right) = (\sigma - 1) \cdot L \left| n_{P;1}, \ldots, n_{P;L} \right), \tag{13}$$

so that all states in the subspace $\mathcal{P}$ transform symmetrically under the action of the global $U(1)$ symmetry generated by $\hat{N}_P + \hat{N}_B$. Furthermore, note that by counting the number of basis states spanning $\mathcal{P}$ it follows that $\dim \mathcal{H} = \dim \mathcal{P}$.

Now, we define the map

$$\begin{aligned} I : \mathcal{P} &\longrightarrow \mathcal{H} \\ \left| \psi \right) &\longmapsto \left| \psi \right\rangle, \end{aligned} \tag{14}$$

identifying states $\left| \psi \right) \in \mathcal{P}$ in the subspace of the doubled system with states $\left| \psi \right\rangle \in \mathcal{H}$ in the original Hilbert space as shown in Fig. 2. Since $g(x)$ is invertible and $\dim \mathcal{P} = \dim \mathcal{H}_P = \dim \mathcal{H}$, it follows that $I$ is invertible. Next, we define the projected purified operator $\hat{O}_{PP} : \mathcal{P} \longrightarrow \mathcal{P}$ by

$$\hat{O} = I \hat{O}_{PP} I^{-1}. \tag{15}$$

Assuming $\hat{O}_{PP}$ exists, this definition implies in particular that

$$\langle n_1, \ldots, n_L | \hat{O} | n'_1, \ldots, n'_L \rangle = (n_{P;1}, \ldots, n_{P;L} | \hat{O}_{PP} | n'_{P;1}, \ldots, n'_{P;L}), \tag{16}$$

that is, the matrix representations of $\hat{O}$ and $\hat{O}_{PP}$ in the local basis sets $\{| n_1, \ldots, n_L \rangle\}$ and $\{| n_{P;1}, \ldots, n_{P;L} )\}$ are identical. We can, hence, work with $\hat{O}_{PP}$ in the subspace $\mathcal{P}$ instead of $\hat{O}$. In order to show that $\hat{O}_{PP}$ always exists, we construct it explicitly. For that purpose, we note that the above definition of $\mathcal{P}$ is equivalent to

$$\left| \psi \right) \in \mathcal{P} \Longleftrightarrow (\hat{n}_{P;j} + \hat{n}_{B;j}) \left| \psi \right) = (\sigma - 1) \left| \psi \right) \quad \text{for all } j \in \{1, \ldots, L\}. \tag{17}$$

But this means that each operator $\hat{O}_{PP}$ has to satisfy

$$\left[\hat{O}_{PP}, \hat{n}_{P;j} + \hat{n}_{B;j}\right] = 0 \quad \text{for all } j \in \{1, \dots, L\}. \tag{18}$$

This motivates us to define balancing operators $\hat{\beta}_{B;j}/\hat{\beta}_{B;j}^{\dagger} : \mathcal{H}_{B,\sigma} \longrightarrow \mathcal{H}_{B,\sigma}$

$$_B\langle n'_{B;j}|\hat{\beta}_{B;j}|n_{B;j}\rangle_B = \delta_{n'_{B;j}, n_{B;j}+1} \tag{19}$$

$$_B\langle n'_{B;j}|\hat{\beta}_{B;j}^{\dagger}|n_{B;j}\rangle_B = \delta_{n'_{B;j}, n_{B;j}-1} \tag{20}$$

$$\left[\hat{\beta}_{B;j}^{(\dagger)}, \hat{b}_{P;k}^{(\dagger)}\right] = 0. \tag{21}$$

Since every operator $\hat{O}_P \otimes \hat{\mathbf{1}}_B$ acting non-trivially only on $\mathcal{H}_P$ can be expressed as function of a product of ladder operators $\hat{b}_{P;j}^{[\dagger]}$, we can thus map it to $\mathcal{P}$ through the transformations

$$\hat{b}_{P;j}^{\dagger} \longrightarrow \hat{b}_{P;j}^{\dagger}\hat{\beta}_{B;j} \quad \text{and} \quad \hat{b}_{P;j} \longrightarrow \hat{b}_{P;j}\hat{\beta}_{B;j}^{\dagger}, \tag{22}$$

and imposing the local gauge fixing conditions Eq. (17). By means of this transformation, which is shown graphically in Fig. 1, the local conservation laws Eq. (18) are fulfilled. Note that $\hat{\beta}_{B;j}^{\dagger}\hat{\beta}_{B;j} \neq \hat{n}_{B;j}$.

There is also another way to introduce projected purified operators. We can define the projection operator

$$\hat{P} = \sum_{\{n_{P;j}\}} \left|n_{P;1}, \dots, n_{P;L}\right)\left(n_{P;1}, \dots, n_{P;L}\right|, \tag{23}$$

and look for operators satisfying $\hat{P}\hat{O}_{PP}\hat{P} = \hat{O}_{PP}$. Those operators are manifestly invariant under a projection into $\mathcal{P}$ and therefore, ignoring zero elements, have the same matrix elements in both $\mathcal{H}$ and $\mathcal{P}$. Here the important observation is that restricting the ansatz class of states $|\psi\rangle_{PB} \in \mathcal{H}_{PB}$ to $\mathcal{P}$, we have found a one-to-one mapping between $\mathcal{H}$ and $\mathcal{P} \subset \mathcal{H}_{PB}$, and the states $|\psi\rangle = \hat{P}|\psi\rangle_{PB}$ transform under the global $U(1)$ symmetry generated by $\hat{N}_P + \hat{N}_B$, obeying Eq. (17).

In the following, we explicitly derive the representation of states in $\mathcal{P}$ in terms of MPS and demonstrate the capability of the introduced $U(1)$ symmetrization to improve the numerical efficiency of MPS calculations. For that purpose, we briefly recapitulate $U(1)$-invariant MPS before digging into the technical details of the projection.

## 5 $U(1)$ Symmetries in Matrix-Product States

$$|\psi\rangle \equiv \cdots M_1 \overset{m_1}{—} M_2 \overset{m_2}{—} \cdots \overset{m_{L-1}}{—} M_L \cdots \rightarrow \cdots M_1 \rightarrow M_2 \rightarrow \cdots \rightarrow M_L \cdots \equiv |\psi\rangle_N$$
$$\sigma_1 \quad \sigma_2 \quad \sigma_L \qquad n_1(\sigma_1) \quad n_2(\sigma_2) \qquad n_L(\sigma_L)$$

Figure 3: Schematic of the tensor network of a MPS. Horizontal lines denote the internal indices with bond dimension $m_j$, whereas the vertical lines denote physical indices with dimension $d$. Dotted lines to the left and right indicate the dummy indices $m_0$ and $m_L$.

Consider a state $|\psi\rangle$ as described in Eq. (9). Within the MPS formulation [31], the coefficients $\psi_{\sigma_1 \dots \sigma_L}$ are expanded into a tensor train of rank-3 tensors $M_{j;m_{j-1},m_j}^{\sigma_j}$. For each lattice

site $j$, there is a set of $\sigma$ matrices $M_j^{\sigma_j} \in \mathbb{C}^{m_{j-1} \times m_j}$. We refer to the matrix dimensions $m_j$ as bond dimensions. A compact representation of $|\psi\rangle$ is then given by

$$|\psi\rangle = \sum_{\sigma_1,\dots,\sigma_L} \underbrace{M_1^{\sigma_1} \cdots M_L^{\sigma_L}}_{\psi_{\sigma_1\dots\sigma_L}} |\sigma_1 \cdots \sigma_L\rangle \,, \tag{24}$$

where neighboring matrices are contracted over their shared bond indices: $M^{\sigma_j} M^{\sigma_{j+1}} = \sum_{m_j} M_{j;m_{j-1},m_j}^{\sigma_j} M_{j+1;m_j,m_{j+1}}^{\sigma_{j+1}}$. Commonly, these contractions are represented pictographically. Each tensor is drawn as a shape with as many legs attached to it as there are indices. Then, contractions over shared indices are indicated by connecting the corresponding legs as shown in Fig. 3 for the case of a MPS.

In order to exploit $U(1)$ symmetries, let us consider a Hamiltonian $\hat{H} : \mathcal{H} \longrightarrow \mathcal{H}$ of a system and $\hat{N} : \mathcal{H} \longrightarrow \mathcal{H}$ an operator generating a global $U(1)$ symmetry of $\hat{H}$, i.e.,

$$\left[\hat{H},\hat{N}\right] = 0, \quad \hat{N} = \sum_{j=1}^{L} \hat{n}_j, \quad [\hat{n}_j, \hat{n}_k] = 0, \tag{25}$$

with local density operators $\hat{n}_j : \mathcal{H}_\sigma \longrightarrow \mathcal{H}_\sigma$ acting only on the $j$th lattice site.

Since $[\hat{H}, \hat{N}] = 0$, we can diagonalize both operators $\hat{H}$ and $\hat{N}$ in the same basis. Let this basis be spanned by $\{|N\rangle\}$ with $\hat{N}|N\rangle = N|N\rangle$ as well as $\langle N|N'\rangle = \delta_{N,N'}$. $N$ is called the global quantum number of the state $|N\rangle$. A state $|\psi\rangle \in \mathcal{H}$ can now be expanded in terms of the simultaneous eigenstates $|n_1,\dots,n_L\rangle \in \mathcal{H}$ of $\hat{N}$ with $N = \sum_j n_j$ and labels $n_j$ denoting the eigenvalues of the local operators[3] $\hat{n}_j$:

$$|\psi\rangle = \sum_{n_1,\dots,n_L} \psi_{n_1\dots n_L} |n_1,\dots,n_L\rangle = \sum_{n_1,\dots,n_L} M_1^{n_1} \cdots M_L^{n_L} |n_1,\dots,n_L\rangle \,. \tag{26}$$

As a consequence of the Wigner-Eckart theorem, it can be shown [35, 36] that the site tensors decompose according to

$$\left(M_{j;\alpha_{j-1},\alpha_j}^{n_j}\right)_{m_{j-1;\alpha_{j-1}},m_{j;\alpha_j}} = M_{j;\underbrace{\alpha_{j-1},m_{j-1;\alpha_{j-1}}}_{a_{j-1}},\underbrace{\alpha_j,m_{j;\alpha_j}}_{a_j}}^{n_j} = T_{j;a_{j-1},a_j}^{n_j} \cdot S_{j;\alpha_{j-1},\alpha_j}^{n_j} \,, \tag{27}$$

with

$$S_{j;\alpha_{j-1},\alpha_j}^{n_j} = \delta(n_j + \alpha_{j-1} - \alpha_j) \,, \tag{28}$$

where we interpret in the following

$$T_{j;a_{j-1},a_j}^{n_j} = \left(T_{j;\alpha_{j-1},\alpha_j}^{n_j}\right)_{m_{j-1;\alpha_{j-1}},m_{j;\alpha_j}} \,, \quad \text{hence} \quad T_{j;a_{j-1},a_j}^{n_j} \in \mathbb{C}^{m_{j-1;\alpha_{j-1}} \times m_{j;\alpha_j}} \,. \tag{29}$$

Here, the indices $\alpha_{j-1}, \alpha_j$ are labeling irreducible representations of the $U(1)$ symmetry on the bond spaces. Hence, we can describe a state by its rank-5 site tensors $M_{j;a_{j-1},a_j}^{n_j}$ and benefit from their block structure. The matrices $M_j^{n_j}$ are decomposed into blocks $T_{j;\alpha_{j-1},\alpha_j}^{n_j}$ with overall dimensions $m_j = \sum_{\alpha_j} m_{j;\alpha_j}$. However, matrix multiplications only scale with the block bond dimensions $m_{j;\alpha_j}$ and are thus cheaper by a factor of $\left(\frac{m_j}{m_{j;\alpha_j}}\right)^3$, i.e., typically $\sim \mathcal{O}(10)-\mathcal{O}(100)$.

---

[3]If the local operators have degenerated eigenvalues, more labels have to be used as a set to identify each state uniquely.

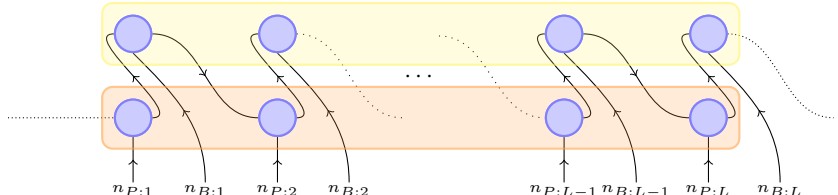

Figure 4: MPS representation in an enlarged Hilbert space with each physical site accompanied by a bath site.

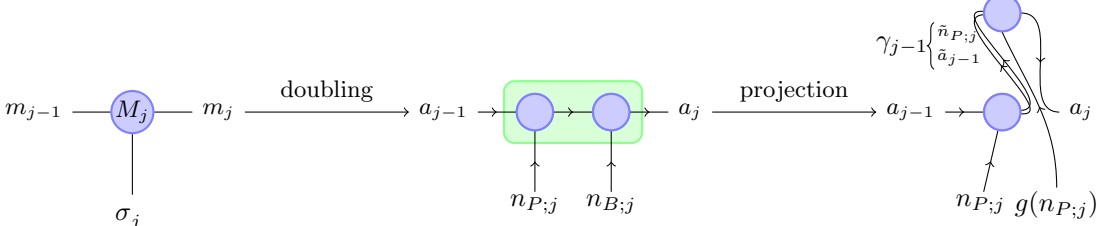

Figure 5: Decomposition of general MPS tensor (left) into $U(1)$-invariant physical and bath-site tensors (center). Projection of the $U(1)$-invariant MPS (center) into the subspace $\mathcal{P}$ (right) enforcing the local gauge condition given in Eq. (17). Decomposition of the introduced auxiliary index $\tilde{a}_{j-1}$ into irreducible representation of the local conservation law generated by $\hat{n}_{P;j} + \hat{n}_{B;j}$ is sketched by double bonds $\gamma_{j-1} \rightarrow (\tilde{n}_{P;j}, \tilde{a}_{j-1})$.

# 6  $U(1)$-Invariant Matrix-Product States with Bath Sites

The introduced mapping from an operator breaking a global $U(1)$ symmetry to one conserving a $U(1)$ symmetry (see Sec. 4) can be exploited to efficiently reduce the matrix sizes of MPS representations. The key observation is that, while purified states in the doubled Hilbert space in general have a huge redundancy that comes with additional gauge degrees of freedom, the projection into $\mathcal{P}$ fixes all these gauge degrees of freedom by the $L$ local gauge constraints given in Eq. (17). Here, we discuss the implications on the projection of purified MPS into $\mathcal{P}$ and an important connection between the Schmidt decomposition of the purified states and the 1RDM. The latter is being derived rigorosly in App. A and also allows to give bounds on the numerical complexity of this mapping when allowing for truncation (App. B). We summarize our findings at the end of this section.

Let again $|\psi\rangle \in \mathcal{H}$ and consider its single-site representation

$$|\psi\rangle = \sum_{m_{j-1}, m_j, n_j} M^{n_j}_{j; m_{j-1}, m_j} |m_{j-1}\rangle \otimes |n_j\rangle \otimes |m_j\rangle \;, \tag{30}$$

with $\langle m_{j-1}|m'_{j-1}\rangle = \delta_{m_{j-1}, m'_{j-1}}$ and $\langle m_j|m'_j\rangle = \delta_{m_j, m'_j}$. Following the previous considerations, we take this state representation into the subspace $\mathcal{P}$ of the enlarged Hilbert space $\mathcal{H}_{PB}$ with $N_P + N_B = (\sigma - 1) \cdot L$. We represent the MPS in $\mathcal{H}_{PB}$ by interpreting the single-site representation of $|\psi\rangle$ as a two-site representation in $\mathcal{H}_{PB}$,

$$|\psi\rangle_{PB} = \sum_{\substack{m_{j-1}, m_j \\ n_{P;j}, n_{B;j}}} M^{n_{P;j}, n_{B;j}}_{j; m_{j-1}, m_j} |m_{j-1}\rangle \otimes |n_{P;j}, n_{B;j}\rangle_{PB} \otimes |m_j\rangle \;. \tag{31}$$

Then, we apply the projection into the subspace $\mathcal{P}$ by enforcing the local gauge condition Eq. (17). Pursuing these two steps at all sites $j \in \{1, \ldots, L\}$, the resulting state representation

is in the subspace $\mathcal{P}$ of the enlarged Hilbert space $\mathcal{H}_{PB}$

$$|\psi) = \sum_{\substack{m_{j-1},m_j \\ n_{P;j},n_{B;j}}} M_{j;m_{j-1},m_j}^{n_{P;j},n_{B;j}} \delta_{n_{B;j},g(n_{P;j})} |m_{j-1}\rangle \otimes |n_{P;j},n_{B;j}\rangle_{PB} \otimes |m_j\rangle \ . \tag{32}$$

Then, the site tensors decompose under the global $U(1)$ symmetry as

$$M_{j;m_{j-1},m_j}^{n_{P;j},n_{B;j}} \equiv M_{j;a_{j-1},a_j}^{n_{P;j},n_{B;j}} = T_{j;a_{j-1},a_j}^{n_{P;j},n_{B;j}} \delta(n_{P;j} + n_{B;j} + \alpha_{j-1} - \alpha_j) \,, \tag{33}$$

where again we combine block and matrix indices $(\alpha_j, m_{j;\alpha_j}) \equiv a_j$ as introduced in Eq. (27). A matrix factorization of the decomposed site tensors $M_j^{n_{P;j},n_{B;j}} = \bigoplus_{N_j} T_{j;N_j}^{n_{P;j},n_{B;j}}$ in each symmetry block $n_{P;j} + \alpha_{j-1} = N_j = n_{B;j} - \alpha_j$ yields the MPS representation of $|\psi)$ in the subspace $\mathcal{P}$ of the enlarged Hilbert space

$$T_{j;N_j}^{n_{P;j},n_{B;j}} \equiv T_{j;\alpha_{j-1},\alpha_j}^{n_{P;j},n_{B;j}} = \sum_{c_{j-1}} T_{j;\alpha_{j-1},c_{j-1}}^{n_{P;j}} T_{j;c_{j-1},\alpha_j}^{n_{B;j}} \tag{34}$$

$$\Rightarrow |\psi) = \sum_{\substack{a_{j-1},n_{P;j}, \\ c_{j-1}}} T_{j;a_{j-1},c_{j-1}}^{n_{P;j}} \delta(n_{P;j} + \alpha_{j-1} - \gamma_{j-1}) |a_{j-1}\rangle \otimes |n_{P;j}\rangle \times$$

$$\sum_{a_j,n_{B;j}} T_{j;c_{j-1},a_j}^{n_{B;j}} \delta(n_{B;j} + \gamma_{j-1} - \alpha_j) \delta_{n_{B;j},g(n_{P;j})} |n_{B;j}\rangle \otimes |a_j\rangle \ . \tag{35}$$

In Eq. (34), we introduce the index $c_{j-1}$ as a result of the factorization in each tensor block. Then, we again employ the notation introduced in Eq. (27) to extend this index to also contain $U(1)$ block labels $\gamma_j$: $(\gamma_j, m_{j;\gamma_j}) \equiv c_j$.

The MPS constructed in this way is shown in Fig. 4 and consists of alternating physical and bath sites, which are labeled by the physical and bath degrees of freedom $n_{P;j}$ and $n_{B;j}$, respectively. The delta function $\delta_{n_{B;j},g(n_{P;j})}$ in the last line of Eq. (35) is again the manifestation of the $L$ gauge-fixing conditions imposed in Eq. (17). It motivates the introduction of the auxiliary $U(1)$ irreducible representation (irrep) labels $\eta_j$ enumerating the irreducible representations of each locally conserved quantity between the physical and bath sites. In this way the $U(1)$-irrep labels $\gamma_{j-1}$ can be decomposed into labels $\gamma_{j-1} \rightarrow (\eta_j, \nu_{j-1})$, which need to fulfill $\eta_j + \nu_{j-1} = n_{P;j} + \alpha_{j-1}$. Note that we focus only on the labels for the symmetry blocks and – for convenience – in the following, neglect the bond dimension $m$, which is part of the label $a$. From the local conservation laws and the gauge fixing, we can furthermore conclude that the bond label $\nu_{j-1}$ has only one non-vanishing block with respect to the global $U(1)$ symmetry, which is characterized by a quantum number $(j-1)\cdot(\sigma-1) \equiv \alpha_{j-1}$. Accordingly, there is only one non-vanishing block $\alpha_j$ to the right of the bath site, which is characterized by a quantum number $j \cdot (\sigma-1) \equiv \alpha_j$. In tensor notation, this can be expressed by a reformulation of the local conservation laws at every site, introducing for brevity $N_j = (\sigma-1)\cdot(j-1)$,

$$\sum_{c_{j-1}} T_{j;\alpha_{j-1},c_{j-1}}^{n_{P;j}} T_{j;c_{j-1},\alpha_j}^{n_{B;j}} \delta_{n_{B;j},g(n_{P;j})}$$

$$= \sum_{\eta_j,\nu_{j-1}} T_{j;\alpha_{j-1},(\eta_j,\nu_{j-1})}^{n_{P;j}} \delta(N_j - \alpha_{j-1}) T_{j;(\eta_j,\nu_{j-1}),\alpha_j}^{n_{B;j}} \delta(N_{j+1} - \alpha_j) \delta_{n_{B;j},g(n_{P;j})} \,. \tag{36}$$

Therefore, we find that there is a unique decomposition of the auxiliary bond label $\gamma_{j-1} = (\eta_j, \nu_{j-1})$ given by identifying $\eta_j \equiv n_{P;j}$ and thus also $\nu_{j-1} \equiv \alpha_{j-1}$. This can be

summarized by decomposing the site tensors as

$$
\sum_{c_{j-1}} T_{j;\alpha_{j-1},c_{j-1}}^{n_{P;j}} T_{j;c_{j-1},\alpha_j}^{n_{B;j}} \delta_{n_{B;j},g(n_{P;j})}
$$
$$
= \sum_{\tilde{n}_{P;j},\tilde{\alpha}_{j-1}} T_{j;\alpha_{j-1},(\tilde{\alpha}_{j-1}\tilde{n}_{P;j})}^{n_{P;j}} T_{j;(\tilde{\alpha}_{j-1}\tilde{n}_{P;j}),\alpha_j}^{n_{B;j}} \delta_{\alpha_{j-1},\tilde{\alpha}_{j-1}} \delta_{n_{P;j},\tilde{n}_{P;j}} \delta_{n_{B;j},g(n_{P;j})} , \tag{37}
$$

which is exemplified in Fig. 5 and presumed from now on. Note that this rather cumbersome notation is important to derive the correct connection between the site tensors $T_{j;\alpha_{j-1},(\tilde{\alpha}_{j-1}\tilde{n}_{P;j})}^{n_{P;j}}$ and the 1RDM. However, in what follows we summarize the results of this discussion in a condensed notation and refer the interested reader to App. A.

Now, we consider the 1RDM, which is the central object of the LBO method [13, 40, 42]. The expectation value of the local density operators in the original Hilbert space can be written in terms of the 1RDM $\hat{\rho}_j = \text{Tr}_{k \neq j} \hat{\rho}$,

$$
\langle \hat{n}_j \rangle = \text{Tr}_j \left\{ \hat{\rho}_j \hat{n}_j \right\} = \sum_{n_j} \langle n_j | \hat{\rho}_j \hat{n}_j | n_j \rangle = \sum_{n_j} \rho_{n_j,n_j} n_j . \tag{38}
$$

Note that the diagonal elements $\rho_{n_j,n_j}$ determine the probability to find $n_j$ particles occupying the $j$th physical degree of freedom. After doubling the system, the diagonal elements of the 1RDM $\hat{\rho}_{P;j}$ for states in a mixed-canonical MPS with center of orthogonality at the physical site $j$ can be written as

$$
\rho_{n_{P;j},n_{P;j}} = \left| T_{j;\alpha_{j-1},(\tilde{n}_{P;j},\tilde{\alpha}_{j-1})}^{n_{P;j}} \delta_{n_{P;j},\tilde{n}_{P;j}} \right|^2 \equiv \left| T^{n_{P;j}} \right|^2 . \tag{39}
$$

Here, the important observation is that the auxiliary bond label $\tilde{n}_{P;j}$ is connected to the label of the physical degree of freedom $n_{P;j}$ by the Kronecker-$\delta$. It is then straightforward to derive an important connection between the occupation probabilities $\rho_{n_{P;j},n_{P;j}}$ of the local degrees of freedom and the Schmidt spectrum $\Lambda_j$ for a cut between the physical and bath site. In particular, in App. A, we show that the singular values $\Lambda_{j;\tau}^{n_{P;j}}$ obtained by factorizing the tensor block $T^{n_{P;j}}$ via a SVD fulfill

$$
\sum_{\tau} \left( \Lambda_{j;\tau}^{n_{P;j}} \right)^2 = \rho_{n_{P;j},n_{P;j}} , \tag{40}
$$

where $\tau$ runs over all singular values in the factorized tensor block $T^{n_{P;j}}$. This relation is the key to understand the numerical behavior of the introduced mapping from an intuitive physical picture. As an example, we assume a system that is characterized by a 1RDM whose diagonal elements $\rho_{n_{P;j},n_{P;j}}$ are sharply peaked around some $n_{P;j} \equiv n_0$. Let us denote the probability to find $n_0$ particles at site $j$ by $\rho_{n_0,n_0} \equiv 1 - \delta$ with some small $0 \leq \delta \ll 1$. Then, Eq. (40) tells us that we can discard all tensor blocks $T^{n_{P;j} \neq n_0}$ while maintaining an approximative description of the quantum state with precision $\delta$. More precisely, if $|\psi)$ is the exact state and $|\varphi)$ the state with all tensor blocks $T^{n_{P;j} \neq n_0}$ discarded, then the Hilbert-Schmidt distance fulfills $(\psi|\varphi) = 1 - \delta$. By choosing the truncation threshold $\delta$ more carefully and allowing for truncations in the tensor blocks, the approximation quality can be improved. Notably, the canonical procedures intrinsic to most of the DMRG algorithms already truncate the site tensors in exactly this way [31], i.e., given a truncation threshold $\delta$, singular values $\Lambda_{j;\tau}^{n_{P;j}}$ are discarded until their summed, squared weight reaches $\delta$.

We investigate the dependency of the probability distributions of the single-site occupations $n_j$ given by the diagonal elements of the 1RDM on the previously described truncation scheme in App. B. For simplicity, we assumed strictly exponentially decaying singular values $\Lambda_{j;\tau}^{n_{P;j}} \sim e^{-\tau A_{n_{P;j}}}$. Interestingly, already in the case of moderately large tensor-block dimensions $m_j \sim O(100)$, we find that the overall increase of the bond dimension between the physical and bath site, compared to the bond dimension in the original system, is practically independent on the exponent $A_{n_{P;j}}$. Moreover, in this regime the growth in the bond dimension decays exponentially with the occupation probabilities $\rho_{n_{P;j},n_{P;j}}$. Combining both results, we find a strong argument that this mapping allows the efficient simulation of systems with large local Hilbert spaces and without global $U(1)$ conservation, if the 1RDM is peaked around some single-site occupation. Numerical simulations and estimations from the exact analysis in App. B showed that, typically, the growth in bond dimension is $\sim \mathcal{O}(1)$ and becomes $\sim 10$ only in drastic situations such as coherent states $\rho_{n_j,n_j} \propto \frac{z^{n_j}}{n_j!}e^{-z}$. Note also that if the state accidentally conserves the global $U(1)$ symmetry, there will be only one non-vanishing tensor block per site and no growth of the total bond dimension at all.

In summary, taking MPS to their projected purified counterparts, we find that the occupation probabilities $\rho_{n_{P;j},n_{P;j}}$, i.e., the diagonal elements of the 1RDM of the physical system, control the numerical efficiency of the state representations. Specifying a certain truncated weight $\delta$ and applying the canonical DMRG truncation scheme then yields an approximation to the 1RDM with an error $\sim \delta$ with respect to the 1-norm. Hence, having quickly decaying occupation probabilities, which is typically the case in physical systems, the projected purification provides an efficient approximation scheme.

# 7 The Holstein Model: Example Calculations

In this section, we provide numerical results for the Holstein model. The Hubbard model with superconducting (SC) terms is discussed in App. C where we focus on some technical issues arising from the anti-commutation relations of the electronic ladder operators.

The Holstein model [48] is given by

$$\hat{H} = -t \sum_j \left( \hat{c}_j^\dagger \hat{c}_{j+1} + \text{h.c.} \right) + \omega_0 \sum_j \hat{b}_j^\dagger \hat{b}_j + \gamma \sum_j \hat{n}_j^f \left( \hat{b}_j^\dagger + \hat{b}_j \right), \tag{41}$$

in which $\hat{c}_j^{(\dagger)}$ denotes spinless fermion annihilation (creation) operators, $\hat{n}_j^f = \hat{c}_j^\dagger \hat{c}_j$ the corresponding particle number operators, and $\hat{b}_j^{(\dagger)}$ the bosonic annihilation (creation) operators. The parameters of this model are the hopping amplitude $t$, the phonon frequency $\omega_0$, and the electron-phonon coupling $\gamma$. Here, the total number of spinless fermions $\sum_j \hat{n}_j^f$ is conserved, while the total number of phonons $\sum_j \hat{b}_j^\dagger \hat{b}_j$ is not. Owing to the fermion-phonon interaction, the number of phonons per lattice site can become very large, rendering this model very challenging for DMRG, in particular in the charge-density wave (CDW) phase at half filling [6, 7], for which we also present some numerical results.

We restore the conservation of the global phonon number by adding balancing operators $\hat{\beta}_{B;j}^{(\dagger)}$, according to the procedure described in Sec. 4. The projected purified Hamilton operator then reads

$$\hat{H}_{PP} = -t \sum_j \left( \hat{c}_{P;j}^\dagger \hat{c}_{P;j+1} + \text{h.c.} \right) + \omega_0 \sum_j \hat{b}_{P;j}^\dagger \hat{b}_{P;j} + \gamma \sum_j \hat{n}_j^f \left( \hat{b}_{P;j}^\dagger \hat{\beta}_{B;j} + \hat{b}_{P;j} \hat{\beta}_{B;j}^\dagger \right). \tag{42}$$

Note that the local phonon-density operators transform as $\hat{b}_j^\dagger \hat{b}_j \rightarrow \hat{b}_{P;j}^\dagger \hat{b}_{P;j}$, which follows directly from the specific definition of the balancing operators in Eqs. (19) and (20).

## Numerical results in the CDW phase

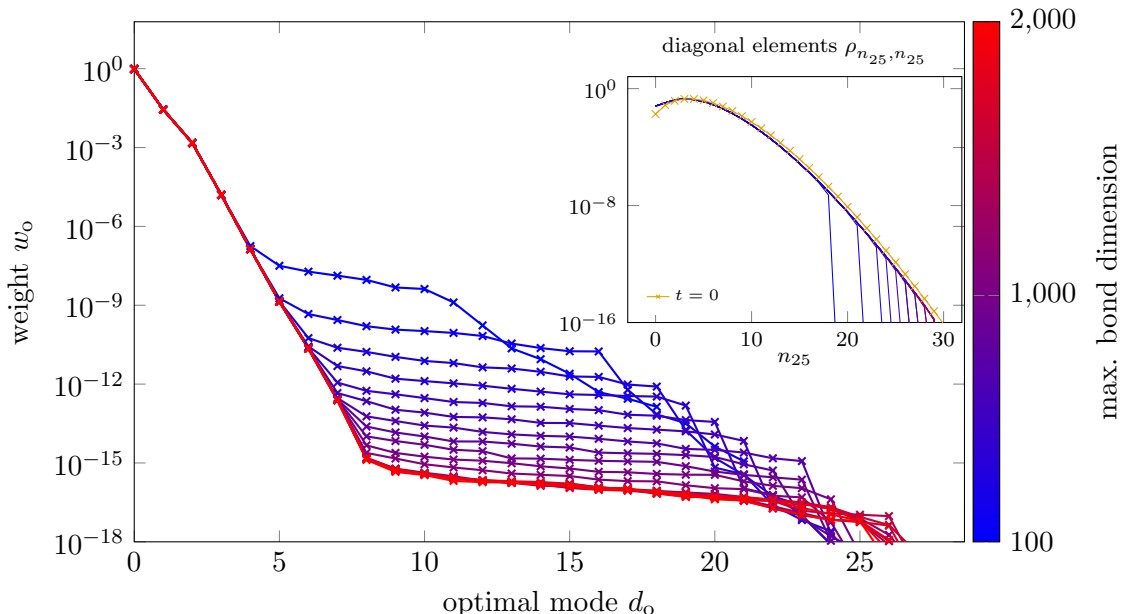

Figure 6: Weight $w_o$ of optimal modes $d_o$ as a function of the maximal bond dimension at the auxiliary bond $\gamma_{25}$ using the projected purification. Data is extracted from the 1RDM $\rho_{n_{25},n'_{25}}$ at the center site ($j = 25$) in the calculated ground state of the half-filled Holstein model with $L = 51$ sites and $N = 25$ fermions, $\omega/t = 1.0, \gamma/t = 2.0$. The inset shows the diagonal elements $\rho_{n_{25},n_{25}}$ indicating the immediate effect of truncations. For comparison, the phonon-excitation probabilities obtained for $t = 0$ are overlayed, indicated by yellow crosses.

In order to illustrate the numerical properties of the mapping introduced in this paper, we performed calculations in the CDW phase of the half-filled Holstein model [7, 40, 49]. This phase is characterized by the formation of bound electron-phonon states (polarons) and a Fermi wave vector $k_F = \pi$, i.e., in a physical picture every second lattice site is occupied by a polaron. In the atomic limit $t \rightarrow 0$, there is an analytic expression for the probability $P_{\text{ph}}(n_j)$ to measure $n_j$ phonons at occupied lattice sites $j$, which is given by

$$P_{\text{ph}}(n_j) = \frac{\gamma^{2n_j}}{\omega_0^{2n_j} n_j!} e^{-\frac{\gamma^2}{\omega_0^2}} . \tag{43}$$

Note that the excitation probabilities are given by the diagonal elements of the 1RDM. Hence, they can be evaluated directly numerically. Another important quantity is the occupation $w_0$ of the optimal modes of the 1RDM $\hat{\rho}_j$, which is also mentioned in App. A. The optimal modes $|d_o\rangle$ are the eigenstates of $\hat{\rho}_j$ and their occupations are the corresponding eigenvalues

$$\hat{\rho}_j = \sum_{d_o} w_o |d_o\rangle \langle d_o| . \tag{44}$$

As discussed elsewhere [13, 40, 42], these constitute an important measure for the quality of the approximation of the phonon states. In our framework, the full 1RDM can be extracted directly

from the projected purified state $|\psi\rangle$ in a mixed canonical representation when contracting physical and bath site tensors $T^{n_{P/B;j}}$ over their auxiliary bond index $\gamma_{j-1}$ (see Eq. (31)):

$$\hat{\rho}_{j;n_j,n'_j} = \text{Tr}_{k\neq j} |\psi\rangle\langle\psi| = \text{Tr}\left\{\left[T^{n'_{P;j}} T^{n'_{B;j}}\right]^{\dagger} T^{n_{P;j}} T^{n_{B;j}}\right\}, \tag{45}$$

where we used the mapping $I$ to identify $n_{P;j} \equiv n_j$ (see also Eq. (51)).

For our calculations, we set $\omega_0/t = 1.0$ and $\gamma/t = 2.0$ so that the model is in the CDW phase. In Fig. 6, the optimal modes of a system with $L = 51$ sites and $N = 25$ fermions are displayed for the ground state and on an occupied lattice site ($j = 25$). The truncation was performed by allowing a maximum discarded weight of $\delta = 10^{-14}$ per auxiliary bond while restricting the total bond dimension to $m_j \leq 2000$. The color-coded graphs correspond to calculations with different, maximally allowed total bond dimensions.

The immediate effect of the truncation on the auxiliary bonds between physical and bath site tensors can be seen as a deviation from the occupation $w_o$ of optimal modes when $w_o$ becomes small. Upon increasing the total bond dimension $m_j$, the distribution $w_o(d_o)$ becomes stationary once $m_j > 1200$. In the inset, the diagonal elements of the 1RDM are shown as a function of $m_j$ and overlayed with the occupation probabilities $P_{\text{ph}}(n_j)$ (Eq. (43)) in the atomic limit. The discarded diagonal elements of $\hat{\rho}_j$ can be deduced from the intersection of the vertical lines with the horizontal axis. Comparing the magnitude at which diagonal elements of $\hat{\rho}_j$ are discarded as a function of $m_j$ to the plateaus of the optimal mode occupation in the main plot, we find a clear correspondence between both. This can be related to the discussion in App. A, we show that with respect to the 1-norm the quality of the approximation of the projected purified state is bounded by the occupation of the optimal modes of $\hat{\rho}_j$, which are not treated correctly. Thus, a scaling analysis in the bond dimension $m_j$ only is sufficient to obtain converged results for the phonon system. Finally, we find that, in accordance with the system being deep in the CDW phase, the diagonal elements $\rho_{j;n_j,n_j}$ are already very close to the excitation probabilities $P_{\text{ph}}(n_j)$ in the atomic limit. Even though the bond dimensions $m_j \leq 2000$ may appear very large, the fact that we are able to exploit global $U(1)$ symmetries for both the fermionic and bosonic system allows us to perform these calculations very efficiently.

We also performed a finite-size scaling of the ground-state energy $E_\delta$ to prove the capability of our approach to deal with large system sizes. Here, we applied a scaling analysis in the numerical precision, tuning the maximal discarded weight per bond from $\delta = 10^{-4}$ to $\delta = 10^{-10}$ and extrapolated $E_\delta$ towards $\delta \to 0$. The number of lattice sites was increased from $L = 51$ sites up to $L = 501$ sites. In Fig. 7, we show the extrapolations and the scaling of the intensive energy density $E_0/L$ as a function of $1/L$. We fit the ground-state energy densities as a function of the number of lattice sites using the ansatz

$$\frac{E_0}{L} = \frac{A}{L} + \varepsilon_\infty. \tag{46}$$

Here, $\lim_{L\to\infty} E_0/L = \varepsilon_\infty$ is the extrapolated ground-state energy density in the thermodynamic limit yielding

$$\varepsilon_\infty = -2.14628340 \pm 4 \cdot 10^{-8}. \tag{47}$$

Note that the given uncertainty is obtained from propagating the errors of the scaling with respect to the discarded weight per bond, which was done for each lattice size $L$. Since bond observables are evaluated with errors whose absolute values are bounded by the discarded weight per bond, this is a numerically exact error bound. Additionally, in the inset of Fig. 7, we plot the total CPU time of a ground-state search running until the convergence threshold $L\delta$ with $\delta = 10^{-8}$ for the relative change in the ground-state energy after a completed sweep was reached. Using two cores of an Intel® Xeon® Gold 6150 CPU @ 2.70GHz, the largest systems with $L = 501$ converged after $\sim 12$ hours.

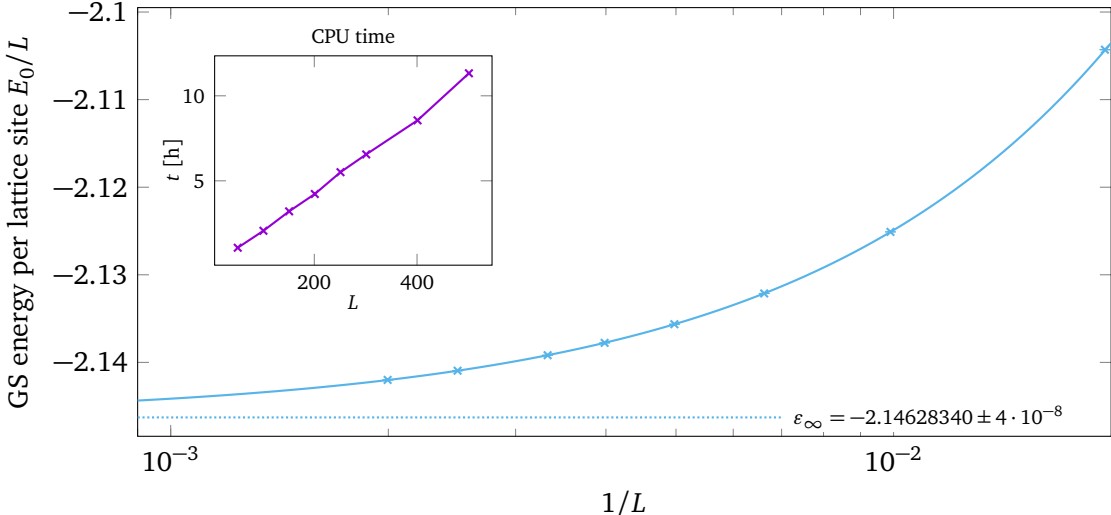

Figure 7: Finite-size scaling for the ground-state energy of the Holstein model at half filling using the projected purification and a two-site DMRG solver. The model is evaluated for parameters $\omega/t = 1.0$, $\gamma/t = 2.0$ at nearly half filling and a maximum discarded weight per bond $\delta = 10^{-10}$. We chose system sizes $L = 51, 101, 151, 201, 251, 301, 401, 501$ and electron fillings $N_{el} = (L-1)/2$. The inset shows the total CPU time for the ground-state search as a function of the number of lattice sites $L$.

# 8 Conclusion

Numerically studying strongly correlated quantum many-body systems with a large number of local degrees of freedom is a challenging problem, in particular for tensor-network methods [6, 7, 40–42]. In this paper we address the problem by introducing a mapping (projected purification) to construct artificial, global $U(1)$ symmetries for models without a generic $U(1)$ symmetry. For any given operator acting on a tensor-product Hilbert space $\mathcal{H}$, we derived a construction scheme that generates its projected purified representation in a subspace of the thermofield doubling of $\mathcal{H}$. We show that both operators can be identified with each other by an isomorphism, but the projected purified representation manifestly conserves global $U(1)$ symmetries. Additionally, we derive a projected purified representation of MPS exploiting the fact that the isomorphism is obtained from a gauge fixing of the additional degrees of freedom introduced by the doubling. Here, the tensors representing the projected purified state can exploit the restored global $U(1)$ symmetry which, for instance, immediately reduces the effective local dimension in each tensor block to 1 providing a significant speedup during numerical calculations when the local Hilbert space dimension is large. We characterize this representation and reveal an intimate relation between the Schmidt values of projected purified MPS and the 1RDM that allows us to estimate the numerical expenses of our representation in comparison to calculations without symmetries.

The mapping into a projected purified representation of operators and states is mostly independent of the underlying implementation. Thereby, it can be used without much effort with already existing toolkits, which we demonstrated by performing numerical calculations [50] on the one-dimensional Holstein model at half filling [7, 10, 49, 51]. The large number of local degrees of freedom that have to be taken into account (we allow up to $N_{ph} = 63$ phonons per lattice site) typically renders large scale calculations very challenging. We perform a finite-size scaling in the CDW phase taking into account a maximum number of $L = 501$ lattice sites

while maintaining a high numerical precision and keeping up to $m_{max} = 2000$ states per bond. Importantly, we showed that convergence in the $U(1)$-symmetry breaking phonon system can be achieved by a scaling in the bond dimension while converging the discarded weight, only. There are no further numerical control parameter, as, for instance, in the LBO, which simplifies both, implementation and numerical simulations.

Due to the reduction of the effective local dimension of the MPS blocks, two-site solvers with a larger numerical complexity can be used [30, 31, 37, 52], as we did in the ground-state calculations of the Holstein model. Therefore, the projected purification allows to apply two-site time-dependent variational principle (2TDVP) [32, 53] as time evolution method to treat systems out of equilibrium. So far, existing methods to tackle such problems mostly [54] use time-evolving block decimation (TEBD) as time stepper, only, due to the high numerical costs when performing two-site updates on systems with a large number of local degrees of freedom [13]. However, TEBD typically requires a much smaller time step to achieve a certain precision, compared to 2TDVP [33]. We thus anticipate that using the presented mapping, out of equilibrium and finite-temperature calculations of such highly complicated systems can become cheaper, more reliable, and straight forward to realize. For instance, we expect this mapping to enable the efficient application of tensor-network algorithms to address questions about lattice electrons coupled to phonons out of equilibrium [22, 55, 56], numerically unbiased. Furthermore, our mapping is compatible with common matrix-product operator (MPO)-based time-evolution methods, e.g., the aforementioned TEBD as well as the MPO $W^{I,II}$ methods [57]. Exhibiting a scaling of the numerical complexity that is at least quadratic in the local dimension [33], these time-evolution schemes should also benefit from taking operators to their projected purified representation.

# Acknowledgements

We thank A. Feiguin, K. Harms, F. Heidrich-Meisner, A. Kantian, R. K. Kessing, and S. R. Manmana for insightful discussions. TK acknowledges financial support by the ERC Starting Grant from the European Union's Horizon 2020 research and innovation program under grant agreement No. 758935. JS and SP were funded by the Deutsche Forschungsgemeinschaft (DFG, German Research Foundation) 207383564/FOR 1807 (projects P4 and P7). SP acknowledges support by the Deutsche Forschungsgemeinschaft (DFG, German Research Foundation) under Germany's Excellence Strategy-426 EXC-2111-390814868. We thank the TU Clausthal for providing access to the Nuku computational cluster.

# A Connection to the Connection to the Single-Site Reduced Density Matrix

The projected purification introduced above is closely related to the 1RDM. We consider the expectation value of the local density operators in the original Hilbert space written in terms of the 1RDM $\hat{\rho}_j = \text{Tr}_{k \neq j} \hat{\rho}$,

$$\langle \hat{n}_j \rangle = \text{Tr}_j \left\{ \hat{\rho}_j \hat{n}_j \right\} = \sum_{n_j} \langle n_j | \hat{\rho}_j \hat{n}_j | n_j \rangle = \sum_{n_j} \rho_{n_j, n_j} n_j \,. \tag{48}$$

Expanding the expectation value of $\hat{n}_{P;j}$ in terms of the physical system's 1RDM $\hat{\rho}_{P;j}$ for states $|\psi\rangle \in \mathcal{P}$ and a mixed-canonical MPS with center of orthogonality at the physical site $j$ yields

$$(\hat{n}_{P,j}) = \text{Tr}_{P;j} \left\{ \hat{\rho}_{P;j} \hat{n}_{P;j} \right\} = \sum_{n_{P;j}} (n_{P;j} | \hat{\rho}_{P;j} \hat{n}_{P;j} | n_{P;j}) = \sum_{n_{P;j}} \rho_{n_{P;j}, n_{P;j}} n_{P;j} \tag{49}$$

$$= \sum_{\substack{n_{P;j}, n'_{P;j}, \\ \tilde{n}_{P;j}, \tilde{\alpha}_{j-1}, \\ \alpha_{j-1}}} n_{P;j} \left( T^{n'_{P;j}}_{j;\alpha_{j-1},(\tilde{n}_{P;j}, \tilde{\alpha}_{j-1})} \delta_{n'_{P;j}, \tilde{n}_{P;j}} \right)^* T^{n_{P;j}}_{j;\alpha_{j-1},(\tilde{n}_{P;j}, \tilde{\alpha}_{j-1})} \delta_{n_{P;j}, \tilde{n}_{P;j}}$$

$$= \sum_{n_{P;j}} n_{P;j} \sum_{\substack{\tilde{n}_{P;j}, \tilde{\alpha}_{j-1}, \\ \alpha_{j-1}}} \left| T^{n_{P;j}}_{j;\alpha_{j-1},(\tilde{n}_{P;j}, \tilde{\alpha}_{j-1})} \delta_{n_{P;j}, \tilde{n}_{P;j}} \right|^2 \,, \tag{50}$$

where we made use of the fact that the local symmetry generators $\hat{n}_{P;j}$ are one-dimensional representations of the local $U(1)$ symmetry (see Fig. 8). From Eq. (16) it follows that Eq. (49) and Eq. (48) are completely equivalent so that

$$\rho_{n_j, n_j} = \rho_{n_{P;j}, n_{P;j}} \,, \tag{51}$$

and thus, comparing to Eq. (50),

$$\rho_{n_j, n_j} = \left| T^{n_{P;j}}_{j;\alpha_{j-1},(\tilde{n}_{P;j}, \tilde{\alpha}_{j-1})} \delta_{n_{P;j}, \tilde{n}_{P;j}} \right|^2 \,. \tag{52}$$

We hence find that the 1RDM of the physical part of $\mathcal{P}$ has the same diagonal elements as the original one. They are given by the trace over the absolute square of the symmetry blocks of the mixed-canonical site tensors. However, the symmetry conservation in $\mathcal{P}$ implies that $\hat{\rho}_{P;j}$ is diagonal whereas $\hat{\rho}_j$ in general is not. We can write the distance with respect to the 1-norm of these two operators by means of the mapping $I$:

$$\|\hat{\rho}_j - I \hat{\rho}_{P;j} I^{-1}\|_1 = \text{Tr}_j \left\{ \hat{\rho}_j \right\} - \text{Tr}_j \left\{ \hat{I} \rho_{P;j} I^{-1} \right\}$$

$$= \text{Tr}_j \left\{ \hat{\rho}_j \right\} - \sum_{n_{P;j}} \left| T^{n_{P;j}}_{j;\alpha_{j-1},(\tilde{n}_{P;j}, \tilde{\alpha}_{j-1})} \delta_{n_{P;j}, \tilde{n}_{P;j}} \right|^2 \,. \tag{53}$$

Here, we link to the LBO method, which expresses $\hat{\rho}_j$ in its eigenbasis (optimal modes) with diagonal elements $w_{n_j}$ so that

$$\|\hat{\rho}_j - I \hat{\rho}_{P;j} I^{-1}\|_1 = \sum_{n_j} w_{n_j} - \sum_{n_{P;j}} \left| T^{n_{P;j}}_{j;\alpha_{j-1},(\tilde{n}_{P;j}, \tilde{\alpha}_{j-1})} \delta_{n_{P;j}, \tilde{n}_{P;j}} \right|^2 \,. \tag{54}$$

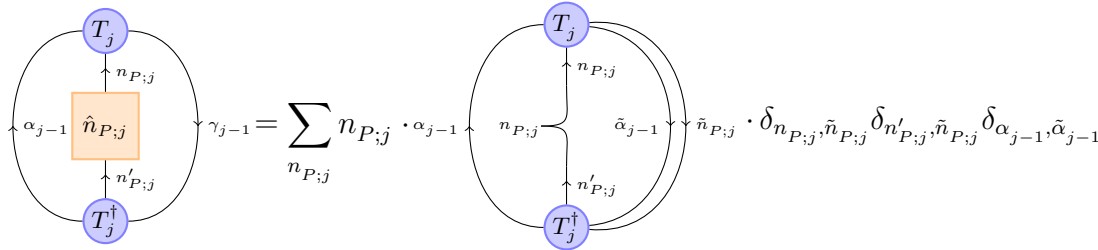

Figure 8: Expectation value of the local density $\langle \hat{n}_{P;j} \rangle$, which by Eq. (52) can be directly related to the diagonal elements of the 1RDM in the eigenbasis $\hat{n}_{P;j}$.

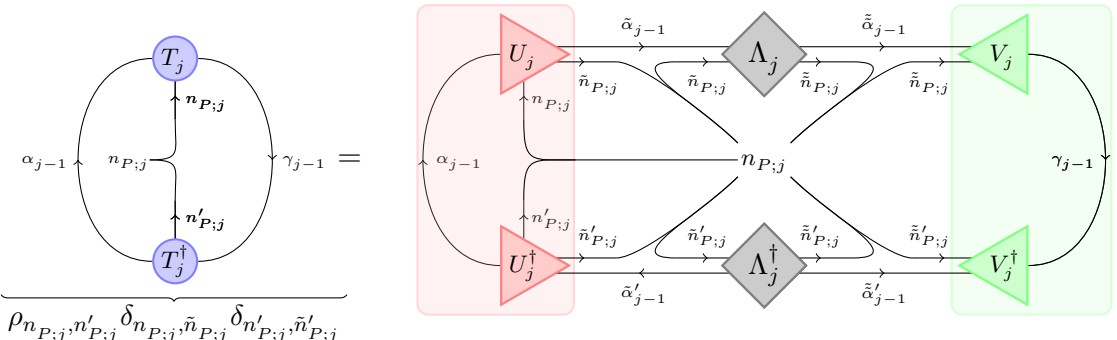

Figure 9: For a given $n_{P;j}$, the diagonal entry of the 1RDM is given by the singular values of the decomposed physical site. Note that we make extensive use of the tensor notation, in particular implicit deltas, which was introduced in [58].

Let us now consider the Schmidt decomposition of a state $|\psi\rangle$ at the auxiliary bond $\gamma_{j-1} = (\tilde{n}_{P;j}, \tilde{\alpha}_{j-1})$. Because $\alpha_j$ is fixed for every $j$, a block for a given $n_{P;j}$ of a physical site can be decomposed individually to

$$T^{n_{P;j}}_{j;\alpha_{j-1},\gamma_{j-1}} = U^{n_{P;j}}_{j;\alpha_{j-1},\gamma_{j-1}} \Lambda^{n_{P;j}}_{j;\gamma_{j-1}} V_{j;\gamma_{j-1}} . \tag{55}$$

The sum over the squared singular values is identified with the corresponding (diagonal) entry of the 1RDM

$$\sum_{\tau} \left( \Lambda^{n_{P;j}}_{j;\gamma_{j-1};\tau} \right)^2 = \rho_{n_{P;j},n_{P;j}} . \tag{56}$$

Note that we implicitly accounted for all constraints arising from the projection into $\mathcal{P}$ and wrote the $\gamma_{j-1}$ on the left only for completeness, as all $\alpha$ are fixed and the $n_{P;j}$ is chosen. In Fig. 9, the argument is given diagrammatically.

Truncating the singular values according to a certain threshold $0 < \delta \ll 1$, so that $\sum_{n_{P;j}} \sum_{\tau} \left( \Lambda^{n_{P;j}}_{j;\tau} \right)^2 < 1 - \delta$ implies a rescaling of the diagonal elements of the 1RDM $(n_{P;j}|\hat{\rho}_{P;j}|n_{P;j})$, which is governed by the decay of the singular values $\Lambda^{n_{P;j}}_{j,\tau}$ in each block. If we assume that the optimal modes of $\hat{\rho}_j$ are truncated in the same way, so that $\sum_{n_j} w_{n_j} < 1 - \delta$, we can compare this expression with Eq. (54). Then, using the invariance of the trace, a truncation of the bond index $\gamma_{j-1}$ by means of the usual MPS truncation routine yields an equivalently precise approximation to $\hat{\rho}_j$ as the truncation occurring in the LBO. In addition, performing the truncation in the projected purified representation automatically favors those eigenvalues of $\hat{\rho}_j$ that have the largest weight without the necessity of constructing the 1RDM

at all. This is an important improvement as it prevents the repeated constructions of $\hat{\rho}_j$ in contrast to the LBO.

# B Characterization of Numerical Expenses

The previous considerations enable us to compare the numerical complexity of typical tensor contractions arising from the MPS representation of states $|\psi\rangle \in \mathcal{P}$ with those of MPS representations without the expansion of the Hilbert space. At first, we point out again that due to the local conservation laws and the gauge fixing, the bond labels $\alpha_{j-1}, \alpha_j$ of the MPS site tensors $T^{n_{P;j}}_{j;\alpha_{j-1},\gamma_{j-1}}$ and $T^{n_{B;j}}_{j;\gamma_{j-1},\alpha_j}$ have only one non-vanishing entry; each of which is given by $\alpha_{j-1} = N_j, \alpha_j = N_{j+1}$ with $N_j$ as defined in Sec. 6. Therefore, without truncation, the bond dimensions $m_{j-1}, m_j$ are identical to those of the site tensors $M^{n_j}_{j;\alpha_{j-1},\alpha_j}$ representing the same state in the physical Hilbert space $\mathcal{H}$ only. There is no additional complexity arising from the representation of $|\psi\rangle \in \mathcal{P}$ on these indices. Furthermore, without truncation the effective bond dimensions on the $\gamma$-bonds are given by $m_{j;\gamma} = n_{P;j} \cdot \min(m_{j-1}, m_j)$. In what follows, we analyze two truncation schemes on these bonds for states in the enlarged Hilbert space $\mathcal{H}_{PB}$. Thereafter, we discuss in which situations these yield a reduced numerical complexity of the most expensive operation during ground-state calculations, i.e., the application of a MPO to a state.

A physically motivated truncation can be defined by exploiting Eq. (40) and discarding all single-site occupations of $\hat{\rho}_j$, whose sum is below a given threshold $\delta > 0$. More precisely, let $\mathcal{D} \subset \left\{0, \cdots, n_{P;j} - 1\right\}$ be a set for which $\sum_{n_{P;j} \in \mathcal{D}} \rho_{n_{P;j}} < 1 - \delta$. Since $\hat{\rho}_j$ is a reduced density matrix, its trace is normalized, and by sorting the diagonal elements such a set can always be defined. Then, all tensor blocks $T^{n_{P;j}}_{j;\alpha_{j-1},\tilde{\alpha}_{j-1}\tilde{n}_{P;j}}$ with $n_{P;j} \notin \mathcal{D}$ are discarded so that the total number of kept states on the auxiliary bond is bounded by $m_{j;\gamma} \leq |\mathcal{D}| \min(m_{j-1}, m_j)$. The physical interpretation is straightforward: All tensor blocks $T^{n_{P;j}}$ that have a negligible single-site occupation $\left|T^{n_{P;j}}\right|^2 = \hat{\rho}_{n_{P;j}}$ are discarded, i.e., empty modes do not contribute to the physics. However, we can give a tighter estimate by considering the explicit distribution of the singular values in each block.

Motivated by the numerical evidence that often the singular values decay exponentially in ground states of one-dimensional (1D) gaped systems [25, 29, 39], we assume such a decay in each block $T^{n_{P;j}}_{j;\alpha_{j-1},\tilde{\alpha}_{j-1}\tilde{n}_{P;j}}$ ($n_{P;j} \in \mathcal{D}$). That means, in the decomposition shown in Fig. 9,

$$\Lambda^{n_{P;j}}_{j;\tau} = e^{-a_{n_{P;j}}\tau}, \quad \sum_{\tau=1}^{m_j} e^{-2a_{n_{P;j}}\tau} = \rho_{n_{P;j}}, \tag{57}$$

for some $a_{n_{P;j}} > 0$ and we abbreviated $m_j \equiv \min(m_{j-1}, m_j)$. Note that $n_{P;j}$ only specifies one block (due to the implicit $\delta_{n_{P;j},\tilde{n}_{P;j}}$) and that we neglected the constant $\alpha_j$ for brevity. Normalization to the single-site occupation yields

$$\rho_{n_{P;j}} = e^{-2a_{n_{P;j}}} \sum_{\tau=0}^{m_j-1} \left(e^{-2a_{n_{P;j}}}\right)^\tau = \frac{e^{-2a_{n_{P;j}}} - e^{-2a_{n_{P;j}}(m_j+1)}}{1 - e^{-2a_{n_{P;j}}}}. \tag{58}$$

Defining $a_{n_{P;j}} = -\frac{1}{2}\log X_{n_{P;j}}$ with $0 < X_{n_{P;j}} < 1$, we can rewrite Eq. (58) into

$$X^{m_j+1}_{n_{P;j}} = X_{n_{P;j}}(1 + \rho_{n_{P;j}}) - \rho_{n_{P;j}}. \tag{59}$$

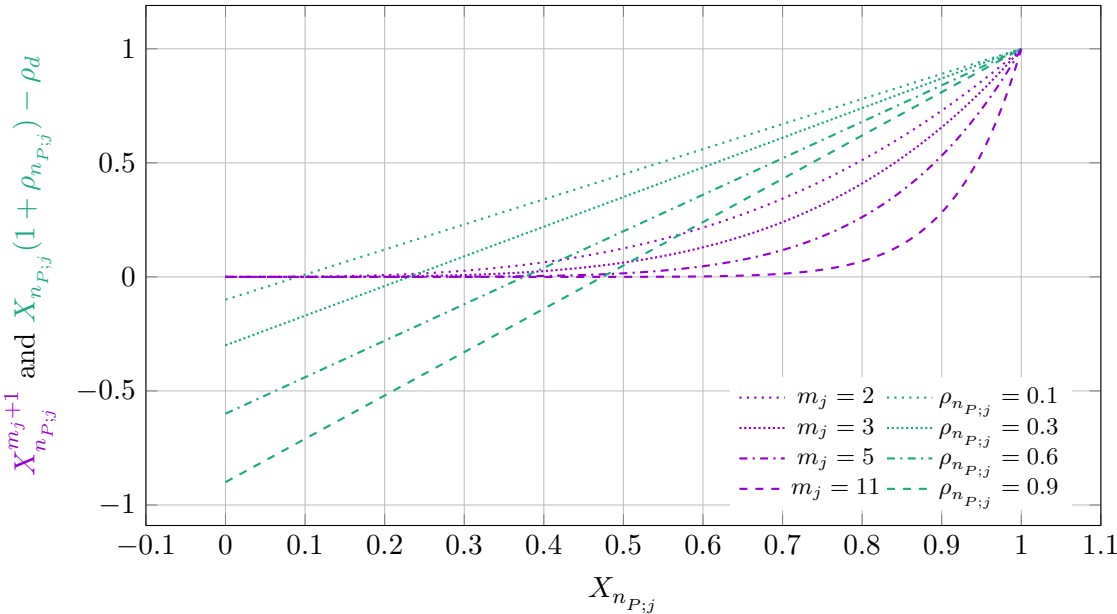

Figure 10: Left (purple) and right (green) hand sides of Eq. (59), $X_{n_{P;j}}$ values at intersections are solutions for distinct pairs of $(\rho_{n_{P;j}}, m_j)$.

Since $\delta \leq \rho_{n_{P;j}} \leq 1$ and $m_j \geq 1$, this equation has only one solution for $X_{n_{P;j}}$ in the given domain, even though there is no closed expression (see Fig. 10 for graphical solution at distinct pairs $(\rho_{n_{P;j}}, m_j)$). Therefore, we consider two limiting cases that yield upper and lower bounds on the decay of the singular values in each tensor block. The lower bound $X_{n_{P;j},\min}$ is obtained through the intersection of the right-hand side with the horizontal axis and can be related to the limit $m_j \gg 1$:

$$0 = X_{n_{P;j},\min}(1 + \rho_{n_{P;j}}) - \rho_{n_{P;j}}$$

$$\Rightarrow X_{n_{P;j}} \geq X_{n_{P;j},\min} = \frac{\rho_{n_{P;j}}}{1 + \rho_{n_{P;j}}} \, . \tag{60}$$

An upper bound $X_{n_{P;j},\max}$ can be established if the right-hand side of Eq. (59) is tangential to the left-hand side

$$\left. \frac{d}{dX_{n_{P;j}}} X_{n_{P;j}}^{m_j+1} \right|_{X_{n_{P;j},\max}} \overset{!}{=} 1 + \rho_{n_{P;j}}$$

$$\Rightarrow X_{n_{P;j}} \leq X_{n_{P;j},\max} = \left( \frac{1 + \rho_{n_{P;j}}}{1 + m_j} \right)^{1/m_j} \, . \tag{61}$$

Combining both bounds, we find

$$-\frac{1}{2m_{n_{P;j}}} \log \frac{1 + \rho_{n_{P;j}}}{1 + m_j} \leq a_{n_{P;j}} \leq -\frac{1}{2} \log \frac{\rho_{n_{P;j}}}{1 + \rho_{n_{P;j}}} \, , \tag{62}$$

which, by introducing normalization constants $A_{n_{P;j},\max/\min}$, limits the decay of the singular values

$$\sqrt{A_{n_{P;j},\min} \left( X_{n_{P;j},\min} \right)^{\tau}} \leq \Lambda_{j;\tau}^{n_{P;j}} \leq \sqrt{A_{n_{P;j},\max} \left( X_{n_{P;j},\max} \right)^{\tau}} \, , \tag{63}$$

and thus can be used to fix upper and lower bounds for the matrix dimensions required on the auxiliary bonds between physical and bath site. The normalization constants are determined from

$$
\rho_{n_{P;j}} = A_{n_{P;j},\eta} \sum_{\tau=1}^{m_j} \left( X_{n_{P;j},\eta} \right)^{\tau} = A_{n_{P;j},\eta} X_{d,\eta} \frac{1 - \left( X_{n_{P;j},\eta} \right)^{m_j}}{1 - X_{n_{P;j},\eta}}
$$
$$
\Rightarrow A_{n_{P;j},\eta} = \frac{1 - X_{n_{P;j},\eta}}{X_{n_{P;j},\eta}} \frac{\rho_{n_{P;j}}}{1 - \left[ X_{n_{P;j},\eta} \right]^{m_j}} , \tag{64}
$$

with $\eta = \min, \max$. We introduce a truncation threshold $\delta'_{n_{P;j}}$ for each block so that for singular values with $\tau \leq m'_{n_{P;j},\eta} \leq m_j$, we obtain

$$
\rho_{n_{P;j}} - \delta'_{n_{P;j}} \geq A_{n_{P;j},\eta} \sum_{\tau=1}^{m'_{n_{P;j},\eta}} \left( X_{n_{P;j},\eta} \right)^{\tau} = \rho_{n_{P;j}} \frac{1 - \left[ X_{n_{P;j},\eta} \right]^{m'_{n_{P;j},\eta}}}{1 - \left( X_{n_{P;j},\eta} \right)^{m_j}}
$$
$$
\Rightarrow \left[ X_{n_{P;j},\eta} \right]^{m'_{n_{P;j},\eta}} \geq 1 - \left( 1 - \frac{\delta'_{n_{P;j}}}{\rho_{n_{P;j}}} \right) \left( 1 - \left( X_{n_{P;j},\eta} \right)^{m_j} \right) . \tag{65}
$$

For this inequality to hold, we necessarily need $\rho_{n_{P;j}} - \delta'_{n_{P;j}} \geq 0$, because $A_{n_{P;j},\eta}, X_{n_{P;j},\eta} > 0$. This is ensured by taking $n_{P;j} \in \mathcal{D}$ and choosing $\delta'_{n_{P;j}} = \max(\frac{\delta}{|\mathcal{D}|}, \min_{n_{P;j} \in \mathcal{D}} \rho_{n_{P;j}})$ as truncation scheme. Then, taking the logarithm of both sides and solving for $m'_{n_{P;j},\eta}$, we divide by $\log X_{n_{P;j},\eta} < 0$ so that

$$
m'_{n_{P;j},\eta} \leq \frac{\log \left\{ 1 - \left( 1 - R_{n_{P;j}} \right) \left( 1 - \left[ X_{n_{P;j},\eta} \right]^{m_j} \right) \right\}}{\log X_{n_{P;j},\eta}} , \tag{66}
$$

where we defined the truncation ratio $R_{n_{P;j}} = \frac{\delta'_{n_{P;j}}}{\rho_{n_{P;j}}} \leq 1$. Imposing equality between the left and right side, we finally obtain an estimation for the upper and lower bounds of the required bond dimension $m'_{n_{P;j},\eta}$ in each block. Introducing the relative change of the number of kept states $F_\eta(m_j, \rho_{n_{P;j}}) = \frac{m'_{n_{P;j},\eta}}{m_j}$, we show the bounds in Fig. 11 for varying $m_j$ and $\rho_{n_{P;j}}$. For the upper bound there are two regimes: In the limit of small truncation ratio $R_{n_{P;j}} \ll 1$ we have $F_{n_{P;j},\max}(m_j, \rho_{n_{P;j}}) \approx 1$, whereas for $R_{n_{P;j}} \rightarrow 1$ there is a sharp drop towards zero. The transition regime between both asymptotics is governed by the physical bond dimension $m_j$ and shifts towards larger values of $\rho_{n_{P;j}}$ as $m_j$ increases. The lower bound exhibits a power-law decay over several magnitudes of $\rho_{n_{P;j}}$ and saturates towards one if $m_j$ is small (Fig. 11). Finally, from Fig. 10 we can deduce that if $m_j \gg 1$, the lower bound becomes an increasingly better approximation for the bond dimension $m'_{n_{P;j},j}$.

In summary, we found that for small physical bond dimension $m_j$ characterizing the approximation of the state without bath sites, the bond dimension $m'_{j,n_{P;j}}$ between physical and auxiliary sites is of the order of $|\mathcal{D}'|m_j$ if $m_j$ is small ($\sim \mathcal{O}(1)$) and $\mathcal{D}' = \left\{ n_{P;j} \mid \rho_{n_{P;j}} > \delta \right\}$. However, if $m_j \gg 1$, the relative value of the bond dimension $m'_{j,n_j^P}$ per tensor block compared

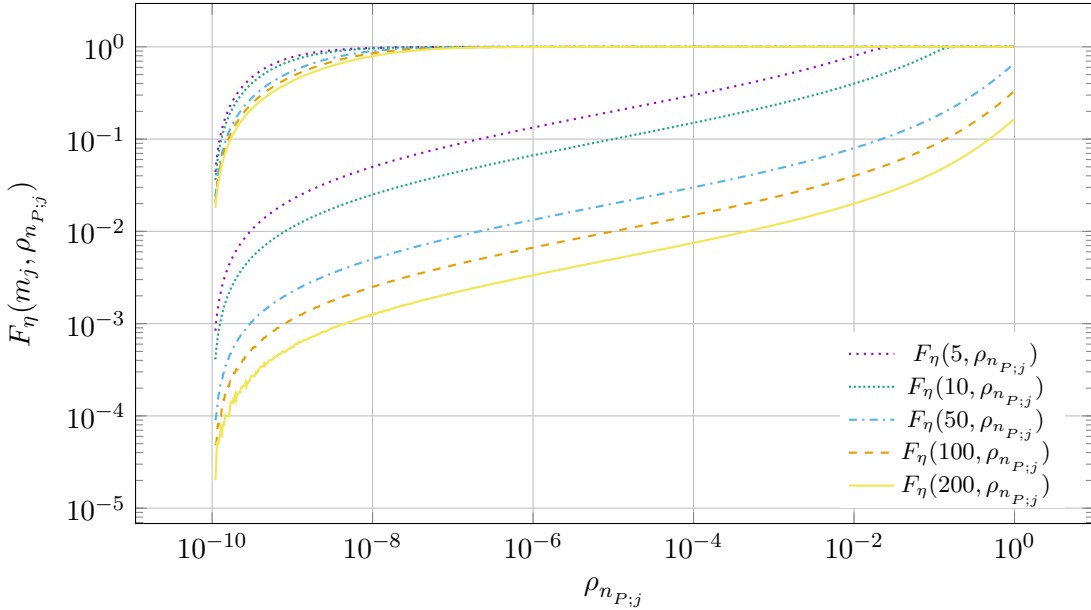

Figure 11: Upper and lower bounds $F_{\text{max/min}}(m_j, \rho_{n_{P;j}})$ for relative change in bond dimension $\dfrac{m'_{n_{P;j}}}{m_j}$ per tensor block on bond between physical and auxiliary sites derived from Eq. (59).

to $m_j$ mostly follows a power law in $\rho_{n_{P;j}}$ and quickly decays to zero. In this situation, the state can be efficiently approximated in the enlarged Hilbert space with a moderate growth of the bond dimension, given that the occupations of the 1RDM $\rho_{n_{P;j}}$ decay fast enough.

In physical problems one is often faced with exponentially decaying occupations of $\rho_{n_{P;j}}$ [59, 60]. Exemplary, we consider a typical, physical bond dimension $m_j = 100$ and assume $\rho_{n_{P;j}} \propto e^{-2n_{P;j}}$ with a truncation threshold of $\delta = 10^{-14}$ and take into consideration a local dimension of $n_{P;j} = 21$ (i.e., permit for 20 occupied states). We use the derived lower bound and obtain $m'_j \approx m_j$. This estimation relies on the assumption of strictly exponentially decaying singular values in each tensor block, which does not necessarily need to be the case in actual calculations. However, a relative growth in the overall bond dimension of $\mathcal{O}(1)$ was also found in our test calculations. Finally, we note that due to the rapid decrease of the lower bound derived above the total local dimension $n_{P;j}$ is not a limiting factor in the first place as long as $m_j$ is large enough. In turn, the decay of the 1RDM occupation strongly dictates the numerical expenses.

We close this appendix by demonstrating the numerical benefits of the above introduced enlargement of the Hilbert space and projection into the subspace $\mathcal{P}$ by considering the scaling of the most expensive calculation in a DMRG two-site ground-state search. This algorithm scales with the application of the MPO to the MPS and has dominating numerical expenses $m_j^3 \cdot w_j \cdot n_{P;j}^2$ if $m_j$ is sufficiently larger than $w_j$. Assuming a typical growth factor 2 between the physical and bath sites, this operation is 8 times more expensive on these bonds than on the original bond between physical sites only. In order to benefit from the introduction of $U(1)$-invariant state representations in the first place, we therefore need to have a reasonably large local dimension $n_{P;j} > \sqrt{8}$, since for $U(1)$-invariant representations all local generators can be chosen as one-dimensional representations. Thus, $n_{P;j} \geq 3$ already speeds up this contraction and the benefits will grow quadratically with larger $n_{P;j}$. We may also consider a decomposition of the MPO bond dimension $w_j$ due to the $U(1)$ symmetry, which typically is

of the order of $2-3$ and thereby also generates an additional speed-up. Finally, we note that the system size is doubled, which could also be incorporated into the estimations. But this is only a constant factor of two and can be compensated easily by the quadratically growing expenses in the local dimension or the decomposition of the MPO bond dimension under the global symmetry.

## C Hubbard Model with Pair Creation and Annihilation

The Hubbard model [61–66] with additional SC terms is given by

$$\hat{H} = -t \sum_{j,\sigma} \left( \hat{c}_{j,\sigma}^\dagger \hat{c}_{j+1,\sigma} + \text{h.c.} \right) + U \sum_j \hat{n}_{j,\uparrow} \hat{n}_{j,\downarrow} + \Delta \sum_j \left( \hat{c}_{j,\uparrow}^\dagger \hat{c}_{j,\downarrow}^\dagger + \text{h.c.} \right), \tag{67}$$

in which $\hat{c}_j^{(\dagger)}$ denotes spin $S = 1/2$ fermion annihilation (creation) operators and $\hat{n}_j = \sum_{\sigma=\uparrow,\downarrow} \hat{c}_{j,\sigma}^\dagger \hat{c}_{j,\sigma}$ the local fermion density operator. The parameters of this model are the hopping amplitude $t$, the interaction strength $U$, and the SC pair creation and annihilation amplitude $\Delta$.

In this model, the pair creation contributions $\propto \Delta$ break the conservation of the global particle number conservation. We restore the corresponding global $U(1)$ symmetry by adding balancing operators $\hat{\beta}_{B;j,\sigma}^{(\dagger)}$ with $\sigma = \uparrow, \downarrow$. The projected purified Hamiltonian now reads

$$\hat{H}_{PP} = -t \sum_{j,\sigma} \left( \hat{c}_{P;j,\sigma}^\dagger \hat{\beta}_{B;j,\sigma} \hat{c}_{P;j+1,\sigma} \hat{\beta}_{B;j+1,\sigma}^\dagger + \text{h.c.} \right) + U \sum_j \hat{n}_{P;j,\uparrow} \hat{n}_{P;j,\downarrow}$$
$$+ \Delta \sum_j \left( \hat{c}_{P;j,\uparrow}^\dagger \hat{\beta}_{B;j,\uparrow} \hat{c}_{P;j,\downarrow}^\dagger \hat{\beta}_{B;j,\downarrow} + \text{h.c.} \right), \tag{68}$$

where local density terms remain unchanged: $\hat{n}_{P;j,\sigma} \hat{\beta}_{B;j,\sigma} \hat{\beta}_{B;j,\sigma}^\dagger = \hat{n}_{P;j,\sigma}$. Exploiting this representation, one of the authors studied the charge-degeneracy points of topologically superconducting islands coupled to normal leads [67–70].

In contrast to the Holstein model, here the projected purification acts on fermions. This causes a subtilty if the fermionic anticommutation relations are implemented in terms of Jordan-Wigner strings [71] as it is usually done, either explicitly or implicitly. For instance, if $\hat{b}_{j,\uparrow}^{(\dagger)}$ are annihilation (creation) operators of hardcore bosons at lattice $j$, then fermionic, bilinear operators can be written in terms of parity operators $\hat{P}_{\hat{b}_{j,\uparrow}}$ as

$$\hat{c}_{j,\uparrow}^\dagger \hat{c}_{j+k,\uparrow} = \hat{b}_j^\dagger \left[ \prod_{l=1}^k \hat{P}_{\hat{b}_{j+l,\uparrow}} \right] \hat{b}_{j+k,\uparrow}. \tag{69}$$

The operator string $\prod_{l=1}^k \hat{P}_{\hat{b}_{j+l,\uparrow}}$ is commonly referred to as Jordan-Wigner string and a consequence of the anticommutation relations. The problem here is that mapping such operator strings into the purified Hilbert space, one has to ensure that they act only in the physical Hilbert space. For instance, if the generation of the anticommutation relations is implemented in the MPS code itself, then typically such Jordan-Wigner strings are created automatically. If this is the case, their effect on the bath sites have to be canceled, which can be done by placing parity operators on bath sites inside the Jordan-Wigner string, for instance,

$$\hat{c}_{j,\uparrow}^\dagger \hat{c}_{j+k,\uparrow} \rightarrow \hat{c}_{j,\uparrow}^\dagger \hat{\beta}_{B;j} \left[ \prod_{l=0}^{k-1} \hat{P}_{\hat{b}_{B;j+l,\uparrow}} \right] \hat{c}_{j+k,\uparrow} \hat{\beta}_{B;j}^\dagger. \tag{70}$$

# D   Object Comparison between LBO and ppDMRG

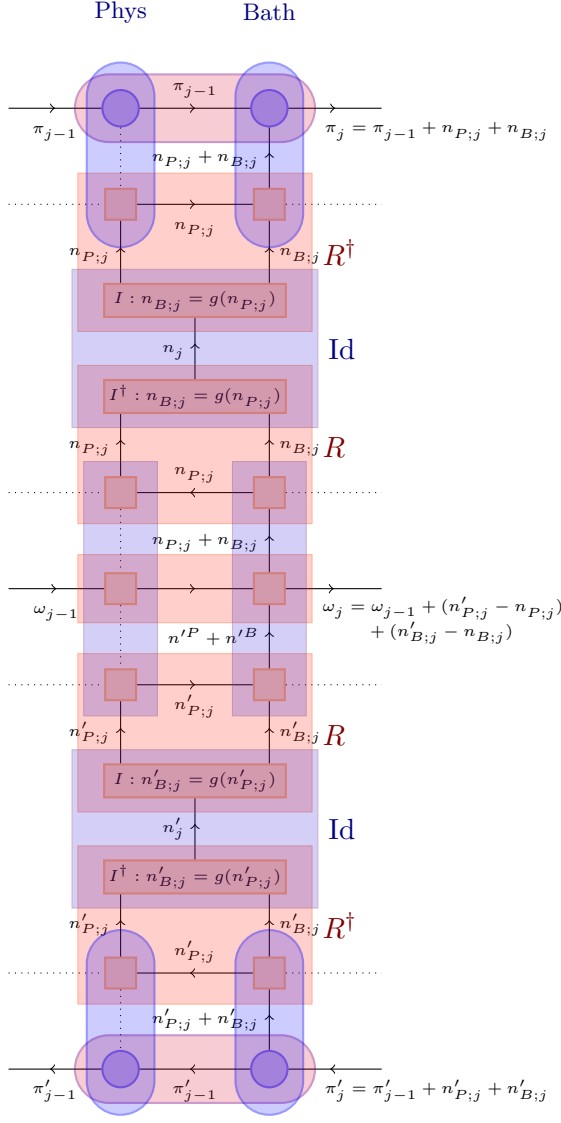

Figure 12: A tensor network representing a single site consisting of an MPS, an MPO, and the adjoint MPS. All tensors are split into several (virtual) objects in order to be rejoined to the tensors used in the LBO (ppDMRG) as highlighted by the red (blue) boxes that contain virtual objects. Note that equivalent bond labels do not indicate the same objects, but only an implicit $\delta$ between the, for brevity not shown, different indicies.

In this appendix, we aim to give an overview of the relationship between the objects used in the LBO and in the ppDMRG. Its main purpose is to support future discussions and developments. It is specifically not intended for implementation purposes, see Sec. 3.

In Fig. 12, a complete sandwich MPS-MPO-MPS for a single site is shown. In order to show the connection between the LBO and the ppDMRG, all tensors are split into virtual objects that are subsequently rejoined in different fashions. On the one hand, the objects coming from the LBO (highlighted with red boxes) are mainly split vertically into parts "belonging" to the physical and the bath Hilbert space. On the other hand, the objects coming from the ppDMRG

(highlighted with blue boxes) needed to be split horizontally so that they could be related to the different objects in the LBO. In particular, the identities containing the maps $I$ and $I^{\dagger}$ do not really appear within the ppDMRG.

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
