# Peer review of "Efficient and Flexible Approach to Simulate Low-Dimensional Quantum Lattice Models with Large Local Hilbert Spaces"

_SciPost Physics, doi:SciPost Phys. 10, 058 (2021)_

## Round 1 · Referee Report · Anonymous (Referee 1) · 2020-12-14

Report

This paper introduces a new numerical method to simulate Hamiltonians with a large local Hilbert space dimension. In particular, it shows how to address Hamiltonians with an operator that explicitly breaks a U(1) number conservation symmetry, for which the faithful description of the ground state requires the inclusion of states with a large particle number per site. This is achieved by mapping the Hamiltonian to one that acts on a different Fock space, in which a different U(1) number conservation symmetry is obeyed. This exploits the block-sparse representations of U(1) symmetric MPS tensors, and provides an efficient method for simulating the desired Hamiltonians.

To the best of my knowledge, the method is original. All calculations (analytical and numerical) seem to be correct and of high quality. The numerical results provided are impressive, and in my opinion achieve the goal that was set forth in the abstract, which is an important problem to address for the simulation of quantum physics.

That being said, there are a few points that I would like to see addressed. I list them in the accompanying sections. Some of them are more pressing than others, and I leave it to the editor to decide which ones are required for publication. My recommendation is to publish after these minor changes.

Requested changes

  1. This is the most pressing point. As the authors note, there already exist two methods for addressing this problem numerically, again using tensor networks, which are called pseudo-site (PS) and the local-basis optimization (LBO). What is the motivation for introducing another one? This is not addressed. It is only noted briefly that there is an advantage over the LBO method in some respect. In my opinion, there needs to be a much more explicit comparison between the methods. Ideally, it should be a direct numerical calculation, but barring that there needs to be at least some rigorous justification given for why the new method is superior to the others. Given that the authors derived such rigorous bounds on the scaling of bond dimension, this should be possible. Or, if the method is not necessarily computationally more powerful than the other two, but it provides a new conceptual framework, this should be explained.

  2. The formulas in the paper are quite hard to follow. This is a combination of the very many explicit indices used, and the small number of tensor diagrams. In order to improve readability, I would recommend that several calculations be moved to the appendix, while being replaced in the main text with intuitive explanations and tensor diagrams. This is particularly true for Section 5, and especially for 5.2, where the key advantages should be explained on physical grounds.

  3. The main idea for the method is quite intuitive, but it is always explained using a very technical language (I didn't understand the concept until reading the formulas). I think it would be very useful to give an intuitive explanation in a couple of sentences, either in the abstract or in the introduction, basically explaining that you can couple each number-changing operator to a bath operator that changes the number in the opposite way, so that the total number is conserved.

  4. Sometimes the notations used in the formulas are not introduced immediately after, creating confusion. Some examples of this are: $\Lambda$ and $V$ in Eq. (42); implicit deltas in Fig. 6; $ w_j$ in the last paragraph of Section 5.2; and $\gamma_{j-1}$ in Eq. (32).

  5. In the last paragraph of Section 5.2, what is the reason for assuming a growth factor of 2 between the physical and bath sites? This statement was a bit confusing to me.

  6. This is just a question: in Eq. (62) is it possible to instead create boson bath sites that couple to the pair creation operator? If not, perhaps the reason for this should be noted.

  • validity: -
  • significance: -
  • originality: -
  • clarity: -
  • formatting: -
  • grammar: -

Author:  Sebastian Paeckel  on 2021-01-14  [id 1152]

(in reply to Report 1 on 2020-12-14)

We thank the referee for her/his positive and constructive report.
We carefully worked through the suggestions and comment in the following on the changes in the order of these.

1. We completely agree with the referee that our approach needs to be put into the context of current, state-of-the-art tensor network techniques used to treat $U(1)$-symmetry breaking models.
In our manuscript, we have added to the appendix a statement that relates an important technical improvement of our method, compared to the LBO:

Importantly, we showed that convergence in the $U(1)$-symmetry breaking phonon system can be achieved by a scaling in the bond dimension while converging the discarded weight, only. There are no further numerical control parameter, as, for instance, in the LBO, which simplifies both, implementation and numerical simulations.

Nevertheless, the main purpose of this manuscript is to introduce this new method and providing a rather complete theoretical formulation of the underlying mapping.
In our opinion, the presentation of the theoretical details of this mapping is already very extensive so that we decided to postpone in-depth numerical comparisons to the PS and LBO methods into a follow-up manuscript (see https://arxiv.org/abs/2011.07412). By that we hope to have given a satisfactory and to some extend complete assessment of our new method.

2. We thank the referee for this valid critique and completely agree that the notational and theoretical examinations are somewhat very exhaustive.
Following the referees suggestion we moved Sec. 5.2 to the appendix and replaced it with a physically motivated reasoning of the observed numerical properties of our mapping.
We also reformulated Sec. 5.1 and simplified the overall notation, e.g., removing the dots indicating the orientation of the tensor legs, as they actually did not serve any purpose anymore.

3. We also thank the referee for this suggestion and extended Sec. 2 presenting the general concept.
We hope that the overall idea is presented in a more transparent manner, now.

4. In order to address the referees comment we have reformulated the corresponding parts, simplified the discussion, and added proper definitions of the bond indices.

5. We thank the referee for raising this interesting question.
In fact, the growth factor of two is a typical value we found during our test calculations.
However, the exact value of the observed growth factor is clearly problem dependent and in situations with very broad distributions of the phonon occupation probabilities we also found larger values.

6. We thank the referee for posing this question.
The balancing operators always have to obey bosonic commutation relations independent on the physical degrees, commutation relations, and statistics.
We hope that the extended Sec. 2 now helps to resolve most of the conceptional questions and possible misunderstandings.

---

## Round 1 · Referee Report · Anonymous (Referee 2) · 2020-12-17

Strengths

  1. A creative solution to a challenging numerical problem.
  2. A high-accuracy large-scale solution of the 1D Holstein model.
  3. The paper is clearly written. a. The main ideas are presented concisely but with sufficient detail b. The mathematical derivations are supplemented with physical motivation.

Weaknesses

  1. The presented method is not used to tackle an outstanding problem but only to refine previous analyses.
  2. A performance comparison between the proposed algorithm and previous schemes (PS and LBO) is missing. Although, this might be redundant given the recent work by the authors (arXiv:2011.07412).

Report

The authors present a novel approach to the numerical solution of low-dimensional quantum many-body problems with a large on-site Hilbert space, as appearing, e.g., in bosonic systems. A naive implementation of DMRG based approaches in such problems has a prohibitively large computational cost that renders the computation impractical. Several past schemes have been suggested to tackle this problem (e.g. PS and LBO). The current progress is achieved by introducing auxiliary degrees of freedom that restore the global U(1) symmetry in an extended Hilbert space. This in turn results in an efficient scheme that can be implemented quite straightforwardly by extending standard MPS codes. The authors make a connection to the LBO approach and flag possible improvements (although not explicitly proven).
Overall, the paper is very clearly written and presents a creative solution to an important numerical problem. For this reason, I believe that the paper meets the acceptance criteria of SciPost and deserves publication.
My only substantial comment is that the proposed method is not used to tackle an outstanding problem.

Requested changes

  1. I suggest to move Section 6.2 to the appendices since no concrete calculation is presented.
  2. A small typo - Above Eq. 23 I believe that "Hamilton" should be "Hamiltonian"

  • validity: high
  • significance: good
  • originality: high
  • clarity: high
  • formatting: excellent
  • grammar: excellent

Author:  Sebastian Paeckel  on 2021-01-14  [id 1153]

(in reply to Report 2 on 2020-12-17)

We thank the referee for her/his positive report on our manuscript and the presented method. We also completely agree with the referee pointing out that our method needs to prove its capability of being able to treat an outstanding problem that could not be dealt with before. However, we believe that in order to exploit the methods capabilities and identify problems to whom it can be applied successfully, at first it is important to understand the theoretical foundations as far as they are accessible to a straightforward analysis. Thus, the main purpose of this manuscript is to present the method with all its technical details and in particular discuss the theoretical implications on the MPS representation and the numerical properties, using projected purified states and operators. Owing to the already very large extend of our discussions we decided to postpone numerically and physically more involved simulations to future work, such as putting the method into the context of current state-of-the-art methods, which we have done in a follow-up study (https://arxiv.org/abs/2011.07412), as already mentioned by the referee.

---

## Round 1 · Referee Report · Anonymous (Referee 3) · 2020-12-21

Strengths

1) the paper introduces a conceptually simple but very powerful approach to numerically treat with problems with boson degrees of freedom using MPS.

2) The paper is quite technical and describes the implementation in great detail, and it is at its core so simple that can readily be implemented within existing codes.

3) The resulting optimization promises to be a game changer; basically a breakthrough that could render previous approaches obsolete and, at the same time, make many problems previously thought unsolvable, suddenly within reach of conventional MPS calculations.

Weaknesses

1) While the concept is simple and straightforward once one understands the premise of the idea, the technical description of the method is so convoluted and cumbersome that it has to be read 20 times before one gets it. The authors introduce notation and symbols that make the reader get lost in it. The dots, tildes, double tildes on top of the indices are terrible choices (particularly the dots... what do they really mean?? The reader finds all the labels with dots all of the sudden and it is not explained. I personally believe that the main idea can be summarized as: a) double the Hilbert space by introducing boson ancillas with the corresponding operators; b) introduce local gauge symmetries (actually local number conservation); c) apply singular value decomposition. That's it. I am convinced that 80% of the technical discussion be moved to an appendix, improving the readability tenfold. Actually, in the end the precise implementation can depend on the practitioner.

Report

The main idea consists of introducing artificial symmetries/conservation laws in an extended Hilbert space. The concept is original and powerful, allowing one to effectively optimize the basis efficiently and at the same time, take advantage of the conservation laws.

As mentioned above, these ideas, as simple as they seem, are a game changer. They potentially represent a big leap that could finally allow one to study problems without global U(1) symmetry/particle conservation, in a numerically efficient way. In a sense, it's like finding that an NP complete problem can be solved in polynomial time. And time will tell if it lives up to the expectations. Results and analysis provided offer convincing evidence in this regard. I recommend it for publication after some changes are considered by the authors.

Requested changes

1 - Simplify the discussion focusing on the conceptual ideas and the intuition behind them. Consider moving the technical details in sections 4 and 5 to appendices.
2 - I beg the authors do improve notation and do something about the dots on top of the indices; are they really necessary? At some point one wonders, if all the labels have a dot, is there a point for them at all? (no pun intended)
3 - A suggestion: Can they use the standard MPS notation with the physical dimension in square brakets? (Maybe they can use curly brakes for the ancillas?)

  • validity: top
  • significance: top
  • originality: top
  • clarity: low
  • formatting: acceptable
  • grammar: excellent

Author:  Sebastian Paeckel  on 2021-01-14  [id 1154]

(in reply to Report 3 on 2020-12-21)

We thank the referee for her/his very positive assessment of our presented method. We agree with the referee's opinion that there were to many notational elements and that a simplification was necessary. As a consequence we removed the dot-notation that, initially, was introduced to differentiate between indices that are contracted and those how were not by indicating the orientation of bonds (ingoing/outgoing <-> right-/left-aligned dots) to remind the reader of the flow of quantum numbers. Since this differentiation is also given via the tildes the dots became pointless.

In order to account for the referee's very constructive suggestions we also modified substantial parts of our manuscript. In more detail we did the following modifications:

1. We extended Sec. 2, now giving a more detailed, conceptual overview of our method and also included a brief, physical motivation about the connection to the single-site reduced density matrix.
We also simplified the former Sec. 5 by moving the detailed discussion in Sec. 5.2 into the appendix and replaced it with a more qualitative and physically motivated discussion in the main text.
We decided to leave Sec. 4 in the main text but simplified the notation (see also the next point).

2. We updated the notation of combined indices labeling irreps and block-dimensions.
For that purpose, we introduced latin letters (e.g., $a_j$) for bonds labeling combined indices of irreducible representations and the corresponding block-dimensions, the first being labeled by the corresponding greek letter (e.g., $a_j \equiv (\alpha_j, m_{j; \alpha_j})$).

In her/his report the referee also suggests to use standard MPS notation with the physical dimension in square brakets. While this is an absolutely valid proposal, we decided to not adopt this notation in our current work. While we are aware of this notation, here, we decided not to adopt it to our current manuscript, as we feel that the introduction of additional brackets may render the notation again more complicated. Instead, we hope that the identification of physical and bath degrees of freedom by the assigned labels ($P$ and $B$) may help the reader to identify the different degrees of freedom when discussing their theoretical relations. However, when it comes to applying our method to physical problems in future works, this definitely is a valuable notational simplification.

---

## Round 2 · Referee Report · Anonymous (Referee 3) · 2021-2-9

Strengths

The new version of the manuscript is considerably improved. The authors now provide a very intuitive and pedagogical introduction with most of the cumbersome technical aspects moved to appendices, and have modified the notation to improve readability.

Report

The authors have addressed all previous comments and suggestions. I recommend it for publications without changes.

---

## Round 2 · List of Changes

1. We moved Sec. 5.2 to the appendix and replaced it with a physically motivated reasoning of the observed numerical properties of our mapping.
  2. We moved Sec. 6.2 into the appendix.
  3. We reformulated Sec. 5.1 and simplified the overall notation, e.g., removing the dots indicating the orientation of the tensor legs.
  4. We extended Sec. 2 presenting the general concept.
  5. We introduced latin letters (e.g., $a_j$) for bonds labeling combined indices of irreducible representations and the corresponding block-dimensions, the first being labeled by the corresponding greek letter (e.g., $a_j \equiv (\alpha_j, m_{j; \alpha_j})$)

---

## Editorial Decision

published